# Partial proteasomal degradation of Lola triggers the male-to-female switch of a dimorphic courtship circuit

Kosei Sato [1], Hiroki Ito[2], Atsushi Yokoyama[3], Gakuta Toba[4] & Daisuke Yamamoto[1,2]

In *Drosophila*, some neurons develop sex-specific neurites that contribute to dimorphic circuits for sex-specific behavior. As opposed to the idea that the sexual dichotomy in transcriptional profiles produced by a sex-specific factor underlies such sex differences, we discovered that the sex-specific cleavage confers the activity as a sexual-fate inducer on the pleiotropic transcription factor Longitudinals lacking (Lola). Surprisingly, Fruitless, another transcription factor with a master regulator role for courtship circuitry formation, directly binds to Lola to protect its cleavage in males. We also show that Lola cleavage involves E3 ubiquitin ligase Cullin1 and 26S proteasome. Our work adds a new dimension to the study of sex-specific behavior and its circuit basis by unveiling a mechanistic link between proteolysis and the sexually dimorphic patterning of circuits. Our findings may also provide new insights into potential causes of the sex-biased incidence of some neuropsychiatric diseases and inspire novel therapeutic approaches to such disorders.

[1] Neuro-Network Evolution Project, Advanced ICT Research Institute, National Institute of Information and Communications Technology (NICT), 588-2 Iwaoka, Nishi-ku, Kobe 651-2492, Japan. [2] Tohoku University Graduate School of Life Sciences, Sendai 980-8577, Japan. [3] Tohoku University Graduate School of Medicine, Sendai 980-8575, Japan. [4] Research Administration/Management Office, University of Tsukuba, Tsukuba 305-8577, Japan. Correspondence and requests for materials should be addressed to K.S. (email: kosei@nict.go.jp) or to D.Y. (email: daichan@nict.go.jp)

Females and males display distinct behavioral patterns. These differences in behavioral patterns are the outcome of the intersex differences in neural circuits[1]. In *Drosophila*, it has well been documented that sexual dimorphisms in neural circuits result, in principle, from sex differences in the number, structure and function of individual cells composing the circuit[2–6]. The development of these cellular sex differences relies primarily on either or both of two sex determination genes, *fruitless* (*fru*) and *doublesex* (*dsx*)[5]. However, it is largely unknown how *fru* and *dsx* produce sex differences in individual cells. We discovered that the *longitudinals lacking* (*lola*) gene is an essential mediator for the *fru* action to masculinize the mAL cell structure. The *lola* locus generates extraordinarily large numbers of mRNA variants (Supplementary Fig. 1), each of which encodes a unique protein isoform. The majority of Lola isoforms share an N-terminal BTB domain implicated in protein-protein interactions[7] and in anchoring ubiquitin proteasome components, followed by an isoform-specific sequence containing a C-terminal zinc finger motif that is a putative DNA-binding region[8]. Interestingly, the *fru* gene product FruM represents another, BTB-zinc finger protein group, which includes a set of male-specific proteins (i.e., FruAM, FruBM and FruEM: nomenclature according to Ref. 9; Supplementary Fig. 1) that function to masculinize certain neurons[10–12] presumably via chromatin remodeling[9]. For example, FruM represses transcription from *roundabout1* (*robo1*), a negative regulator gene for neuritogenesis, thereby allowing male-specific neurite formation in males[13]. Here we demonstrate that the male-biased Lola29M isoform forms a complex with FruBM. We further demonstrate that Lola29M is a precursor of the female-specific Lola29F isoform, an N-terminal truncation product of Lola29M that counteracts Lola29M action so as to inhibit male-specific neurite formation in females. Surprisingly, the male-specific transcriptional repressor FruBM protects Lola29M from its N-terminal truncation, which is mediated by the E3 ubiquitin ligase Cullin1 (Cul1) and 26S proteasome. Thus, by making the masculinizing protein Lola29M resistant to degradation, the male-specific transcription factor FruBM prevents the production of the feminizing protein Lola29F. As a consequence, the male program for dimorphic circuit formation turns on and the female program turns off in the male brain.

In humans, sexually biased incidence of certain neurological disorders has been well recognized, yet the origin of such sex differences remains largely an enigma. For instance, the male-to-female incidence ratios have been reported to vary from 1.37 to 3.7 in Parkinson's disease, the etiology of which likely involves mitochondrial dysfunction due to defects in the proteasomal degradation system[14]. Our finding in *Drosophila* that the neuronal sex-type specification involves proteasomal protein processing will shed light on the hitherto unknown mechanistic link among posttranslational protein modification, neural sex differentiation and complex neurobehavioral traits under normal and disordered conditions.

## Results

### *lola* as a phenotypic modifier of *fru*.

To obtain insights into the mechanism of *fru* actions on neural sex-type specification, we here screened for *fru* modifier genes. In this screen, we took advantage of a gain-of-function effect of *fru*+ to disrupt the compound eye structure when overexpressed in the developing eye disc. Genome-wide searches for genes that can modify the *fru* eye phenotype were conducted by the Gene-Search (GS) system[15]. In this system, a *P*-element vector carrying the GAL4-responsive DNA sequence *UAS* (the *GS*-element) was randomly inserted into the genome of a fly so that transcription units that happened to flank the *GS*-element insertion could be transcribed

in the presence of an arbitrarily chosen *GAL4* transgene (Supplementary Fig. 2a). This study used *GMR-GAL4* to drive transcription via the *GS*-element as well as *UAS-fruB*+ in the developing eye disc (Supplementary Fig. 2b), yielding several enhancers of the *fruB*+-induced eye phenotype, which included *lola* (Supplementary Fig. 2c-e; for other *fru* modifiers see Ref. 16). Conversely, a loss-of-function *lola* mutation dominantly suppressed the *fru*-induced distortion of the eye (Supplementary Fig. 2f,g). In subsequent analyses, we will focus on *lola* because: (1) it encodes proteins of the BTB-zinc finger superfamily to which Fru also belongs and (2) its functions in neurite guidance have been well established[17]. Indeed, we found that reduced courtship toward a female in *fru* hypomorphic males was dominantly enhanced by two different *lola* null alleles (*lola*03089 and *lola*ORE76), which are both homozygous lethal[18,19] (Fig. 1a), and *lola* knockdown *per se* attenuated male courtship activities (Fig. 1b). These results implicate *lola* in the *fru*-dependent formation of courtship circuits.

### Lola isoforms dedicated to sexual differentiation.

We suspected that Fru (more specifically, FruBM, see below) might affect expression of *lola* to exert its neural masculinizing effect in, for example, sexually dimorphic mAL interneurons in the brain (Supplementary Fig. 2h), although no sex differences in the structure or expression of Lola isoforms has been reported and despite the numerousness of Lola isoforms identified to date[20]. To obtain hints as to which of the Lola isoforms might have a role in the *fru*-dependent sexual differentiation, we examined possible effects of isoform-specific knockdown for isoforms 11, 17, 22, 26, 28 and 29, for which *UAS-RNAi* transgenic strains were publicly available, and found that isoforms 22 and 29 interfered with the sex-specific development of *fru*-expressing neurons (see below). Our extensive analyses of Lola expression with an antibody raised against an exon29-specific sequence (anti-Lola-exon 29; Supplementary Fig. 3) revealed two isoforms, Lola29M of ~110kD and Lola29F of ~80kD, the relative amounts of which were different between the sexes in wandering stage third instar larvae (Fig. 1c). The isoform containing the sequence encoded by exon 29 was formally designated as type-Q[20]. We found that Lola29M is more abundant in males than females whereas Lola29F is detectable only in females (Fig. 1d). Immunostaining of the wandering-stage larval CNS with the anti-Lola-exon 29 antibody revealed that Lola29M/F expression is confined to differentiating neurons, including those expressing FruM, but not detected in neuroblasts (Supplementary Fig. 4). The Lola29M and Lola29F isoforms shared exon29, which was connected to the same 5' exons, and transcribed from the same promoters. Because no sex difference was detected in the mRNA species with exon 29 by 5' and 3' RACE (Supplementary Fig. 5), the size difference between the Lola29F and Lola29M proteins was considered to result from posttranslational modification of a protein product. Intriguingly, *fru* mutant (*fruM*−) males that are null for male-specific FruM proteins expressed the female-specific Lola29F isoform with a concomitant reduction in Lola29M (Fig. 1c, d). In contrast, females sexually transformed into males by the *transformer1* (*tra1*) mutation lost Lola29F with an elevated level of Lola29M (Fig. 1c, d). These results imply that the posttranslational modification of the Lola protein is under the control of the sex-determination cascade downstream of *tra* and *fru*.

### Female-specific Lola29F is a truncation product of Lola29M.

To examine whether female-specific Lola29F is a truncation product of male-enriched Lola29M, we attempted to determine the structures of Lola29F and Lola29M and the mechanism whereby these two isoforms are produced in a sex-dependent

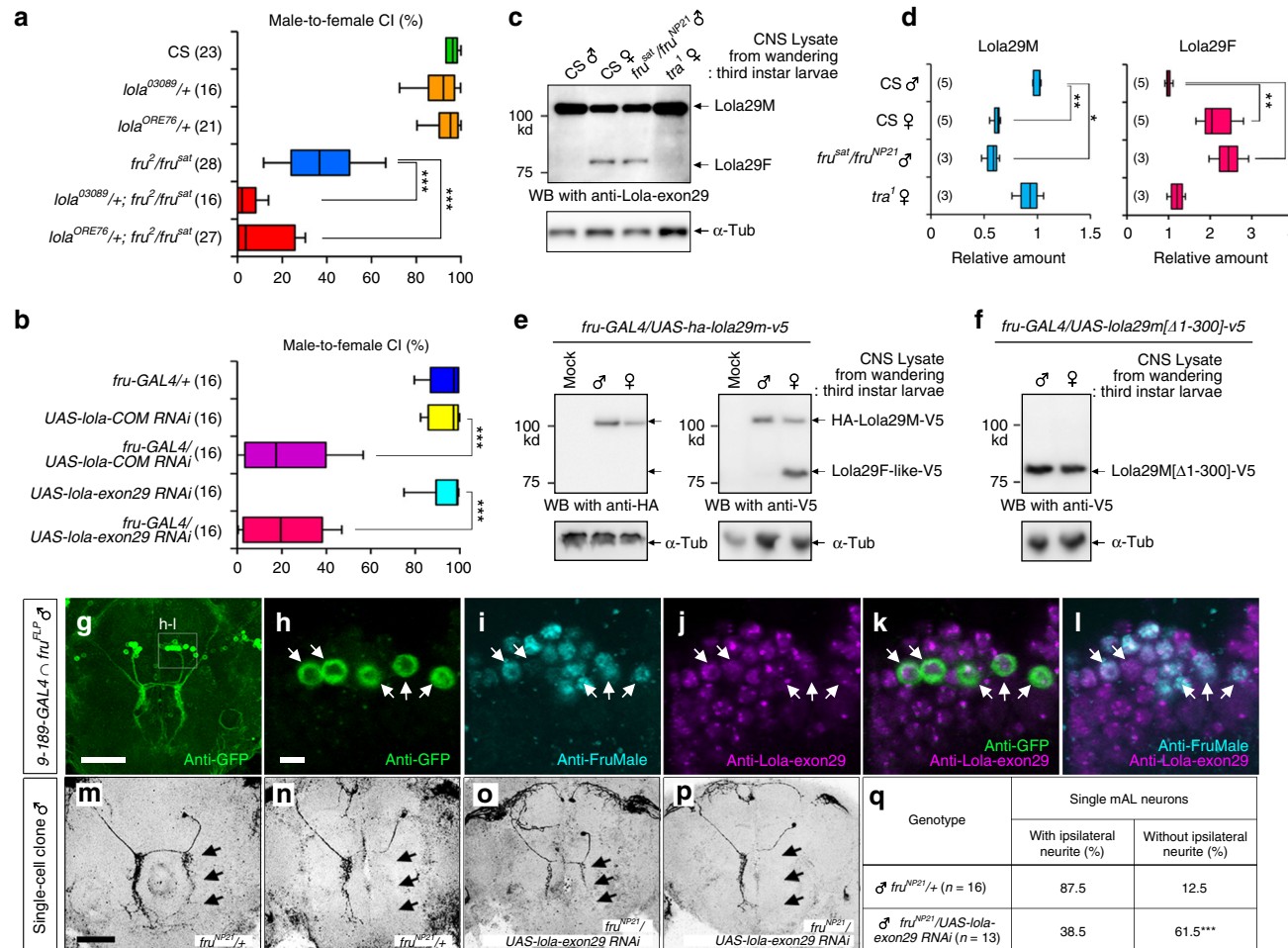

**Fig. 1** *lola* exon 29 contributes to a set of Lola isoforms differentially expressed between females and males. **a**, **b** Male courtship activities. Courtship defects in *fru* hypomorphic males were enhanced in the *lola* heterozygous background (**a**) and knockdown by *lola-COM RNAi* (3rd bar from the top) or by *lola-exon 29 RNAi* (5th bar from the top) suppressed male-to-female courtship (**b**). ***$P < 0.001$ by the Kruskal–Wallis analysis of variance followed by Steel–Dwass post hoc test (**a**, **b**). **c** Western blotting with anti-Lola-exon 29 of CNS extracts obtained from wild-type (CS) males, wild-type females, $fru^{sat}/fru^{NP21}$ males and $tra^1$ homozygous females. **d** Quantification of Lola29M (left-side panel) and Lola29F (right-side panel), as normalized by the values for the wild-type male. The number of replicates is indicated in parentheses. **$P < 0.01$; *$P < 0.05$ by the Kruskal–Wallis analysis of variance followed by Steel's nonparametric multiple comparisons. The box plot shows the median and 10th, 25th, 75th, and 90th percentiles. **e** Overexpression of a Lola29M-encoding sequence decorated with a HA-tag at the N-terminus and a V5-tag at the C-terminus (HA-lola29m-V5) yielded the Lola29M protein in males and less abundantly in females, as well as a Lola29F-like protein in females. Shown is a western blot of larval CNS extracts probed with anti-V5 that recognizes the C-terminus of Lola29M. **f** Lola29M[Δ1-300] lacking a.a. 1–300 resulted in a protein, Lola29F-like, with a molecular weight close to that of Lola29F in both sexes. α-Tubulin (α-Tub) was used as a loading control. **g–l** mAL neurons in the male brain triply stained with antibodies against GFP (**g**, **h**, **k**), FruM (**i**, **l**) and Lola29M/F (**j–l**). GFP was preferentially expressed in mAL neurons by the intersection of $fru^{FLP}$ and *9-189-GAL4*. Scale bars: 50 μm (**g**) or 10 μm (**h**). **m–q** Analysis of single-cell mAL clones that express *lola-exon29 RNAi*. Examples of single-cell clones (**m–p**) and the proportion of flies that carried clones with or without the ipsilateral neurite (**q**). Scale bar: 50 μm. ***$P < 0.001$ by the Fisher's exact probability test. Source data are provided as a Source Data file

manner. We presumed that Lola29F might be a proteolytic product of Lola29M. To test this idea, we overexpressed a *lola29m* gene decorated with an N-terminal HA tag and a C-terminal V5 tag in flies under the control of *fru-GAL4*, and analyzed the lysates from CNS cells by western blotting with antibodies against HA and V5 (Fig. 1e). Remarkably, the anti-V5 antibody detected a shorter band similar to Lola29F (referred to as Lola29F-like hereafter) in addition to the full-length Lola29M, whereas the anti-HA antibody detected only the latter (Fig. 1e). Therefore, the difference between the two isoforms must reside in their N-termini. Overexpression of a series of N-terminal deletants of *lola29m* in *fru-GAL4*-positive cells resulted in the expression of mutant Lola29 proteins; deletion of the N-terminal 300 (Δ1-300) residues yielded a single band of a size similar to that of Lola29F

on western blots (Fig. 1f). We infer that Lola29F is produced by the truncation of Lola29M somewhere between residues 250 and 300 of Lola29M, and later analyses by Edman degradation revealed that this is indeed the case (see below).

Because *lola* plays an important role in neurite patterning, the effects of *lola* knockdown on courtship behavior (Fig. 1b) likely result from a disturbance in the formation of courtship circuits. It has been established that FruM plays a key role in the courtship circuit formation in the male CNS, and some of the *fru* mutant defects in mating behavior have been ascribed to impairments of sex-specific development of *fru*-expressing neurons[10–12]. Therefore, to examine the possible role of Lola29F/M on the courtship circuit formation, we next examined *fru*-expressing mAL interneurons, which exhibit remarkable sex-differences

(Supplementary Fig. 2h), which have been shown to play a role in the processing of sex pheromone information[21–23]. One of the prominent sex differences found in mAL neurons is the absence (in females) or presence (in males) of the ipsilateral neurite. In a subsequent analysis, therefore, we focused on the male-specific ipsilateral neurite formation, where male-specific FruM directly represses transcription of *robo1* in males, thereby allowing mAL neurons to extend the male-specific neurite[13]. Gain of the male-specific ipsilateral neurite (i.e., masculinization) in females or *fruM*-null mutant males can be assessed in neuroblast clones that contain up to 5 mAL neurons, because none of the mAL neurons in control females and control *fruM*-null males possesses this neurite. Loss of the male-specific neurite (i.e., feminization) in males, in contrast, is only detectable in single-cell clones, because any remaining ipsilateral neurite of unaffected cells hinder the neurite loss from affected cells[13]. We therefore examine neuroblast clones for quantifying the masculinizing effect of genetic manipulations and single-cell clones for the feminizing effect.

We first confirmed the immunoreactivity of mAL neurons to the anti-Lola-exon 29 antibody (Fig. 1g-l). We then tested the effect of *lola29* knockdown in these cells. Notably, the expression of RNAi targeting *lola-exon29* markedly increased the formation of mAL neurons without the male-specific ipsilateral neurite in male mAL neurons (Fig. 1m-q). It is noteworthy that the *lola-exon29* RNAi used here effectively inhibited Lola29M production in flies, which retained a high level of expression of other *lola* isoforms (Supplementary Fig. 6).

To further clarify roles of Lola29M and Lola29F in the sex-type specification of neurites, we first wanted to define their molecular structures and the mechanism whereby two Lola29 forms are produced. To determine the exact site of truncation in producing Lola29F from Lola29M, we employed Edman degradation analysis (Fig. 2a). To obtain a large amount of Lola29F sufficient for amino acid sequencing, we overexpressed *lola29m-6xV5* under the control of *fru-GAL4* in female larvae, from which the CNSs were dissected to extract proteins for immunoprecipitating *lola29m-6xV5* products. The band representing Lola29F on the PVDF membrane visualized with Coomassie brilliant blue (CBB; Fig. 2a) was cut out and analyzed, allowing us to determine that the N-terminal five residues were SSTAA (Fig. 2b). This five-residue sequence was uniquely identified as a.a. 264–268 in Lola29M. We therefore conclude that a.a. 1–263 of Lola29M are degraded to yield Lola29F.

In a separate experiment, we detected Lola29F-like, in addition to the expected Lola29M, in S2 cells that had been transfected with *lola29m* alone without a *lola29f* minigene when lysates were prepared with a sufficient delay (~2 days) after transfection (Supplementary Fig. 7). Importantly, the production of Lola29F-like in S2 cells was inhibited by the administration of a ubiquitin-proteasome inhibitor, MG-132 (Fig. 2c) or Lactacystin (Fig. 2d). This result suggests that Lola29F-like is produced by ubiquitin-mediated proteolysis of Lola29M in S2 cells.

Polyubiquitination of ubiquitin at lysine 48 (K48) is known to direct substrate proteins to proteasome-mediated degradation[24]. We found that a K48-linkage-specific polyubiquitin antibody coimmunoprecipitates Lola29M/F in lysates from S2 cells transfected with *lola29m*, and *fruBM* cotransfection markedly diminishes the yield of immunoprecipitates (Supplementary Fig. 8). This result is taken as evidence that Lola29M is indeed ubiquitinated in the absence of FruM.

Ubiquitination takes place at lysine (K) residues. Lola29M has 40 lysine residues, and 15 of them are within the N-terminal 263 amino acid region, which is likely removed in Lola29F (Fig. 2e). We replaced these 15 lysine residues with arginine (R) one by one to examine whether any of these mutations confer to Lola29M the

resistance against proteolysis. Notably, four mutants, K41R, K44R, K47R and K67R, showed a marked reduction in the Lola29F-like production (Fig. 2f). The K41R mutation similarly protected Lola29M from proteolysis in vivo (Fig. 2g). Taking all these results into account, we conclude that Lola29F is a proteolytic product of Lola29M (Fig. 2h).

**Lola29M masculinizes whereas Lola29F feminizes the neurons.** Based on knowledge of their molecular properties, we attempted to determine the roles of Lola29M and Lola29F on sex-specific neurite identity. We found that overexpression of Lola29F-like inhibits the ipsilateral neurite formation; the proportion of single-cell clones without the ipsilateral neurite was significantly increased upon Lola29F-like overexpression in males (Fig. 3a-c). However, mAL neurons with the ipsilateral neurite were still produced, though at a reduced rate, in male flies with Lola29F-like overexpression, implying that an additional, Lola29F-resistant mechanism operates for the ipsilateral neurite formation.

We next examined the effect of overexpression of Lola29M (truncation-resistant Lola29M[K41R]) and Lola29F-like (N-terminally truncated Lola29M[Δ1-300]) in females (Supplementary Fig. 9) and *fruM*-null mutant males (Fig. 3d–h), which lack endogenous FruM that could potentially complicate the analysis. Overexpression of Lola29M induced the male-specific ipsilateral neurite in some females (Supplementary Fig. 9b) and some *fru* mutant males (Fig. 3e), which otherwise lack it without exception (Fig. 3d and Supplementary Fig. 9a). It is conceivable that endogenous Lola29F present in females and *fru* mutant males (see Fig. 1c) hampered the masculinizing action of overexpressed Lola29M[K41R], such that formation of the ipsilateral neurite in mAL neurons was observed in only 20-40% of flies examined (Fig. 3h and Supplementary Fig. 9d). This notion was supported by the observation that the ability of Lola29M[K41R] to restore the male-specific neurite in *fru* mutant males was canceled out by the additional overexpression of Lola29F-like (Fig. 3g). In contrast, overexpression of Lola29F-like alone via an N-terminally deleted *lola29m[Δ1-300]* transgene in females (Supplementary Fig. 9c) or *fru* mutant males (Fig. 3f) had no effect on the mAL neurite structure, which continued to be the female-type in all flies examined.

To examine whether Lola29M and Lola29F are involved in the neural sexual differentiation of neurons other than mAL, we overexpressed truncation-resistant Lola29M or Lola29F-like in *fru*-expressing pheromone receptor neurons (Fig. 4a–g), the central projection of which is sexually dimorphic; male afferent fibers cross the midline (Fig. 4a, b) whereas female counterparts terminate in the ipsilateral neuromere in the prothoracic ganglion[25]. *fru* hypomorphic mutant (*fru2/frusat*) males have a reduced midline crossing (Fig. 4c), offering a sensitized genetic condition in which subtle phenotypic changes can be unambiguously detected[9]. We found that truncation-resistant Lola29M significantly increased (Fig. 4d) and Lola29F-like significantly decreased (Fig. 4e) the midline crossing of sensory axons (Fig. 4g), suggesting that Lola29M and Lola29F contribute to certain dimorphic features of different types of *fru*-expressing neurons. We conclude that Lola29M promotes the male-typical neurite formation and Lola29F counteracts the action of Lola29M.

**Lola29F and Lola29M are required for mating behavior.** To evaluate the contributions of Lola29M and Lola29F to the execution of mating behavior, we measured the courtship index for male courtship vigor and the time to copulation for female receptivity in flies which expressed *lola exon 29-RNAi* together with *lola29m[K41R]* encoding truncation-resistant Lola29M or

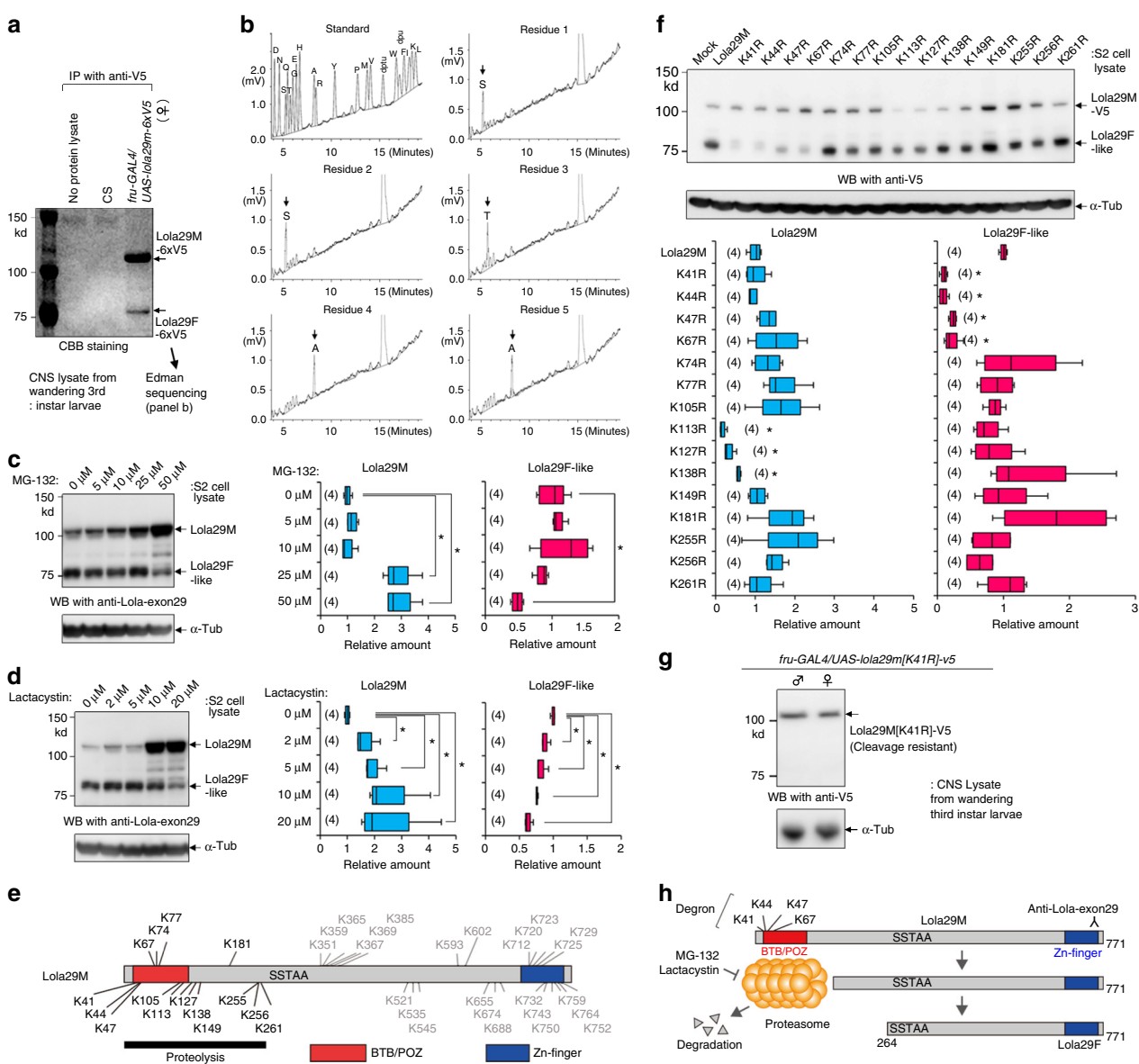

**Fig. 2** Proteasome involvement in the production of Lola29F. **a** A coomasie brilliant blue (CBB) stained PVDF membrane showing results from the biochemical purification of Lola29M and Lola29F. *lola29m-6xv5* was overexpressed *via fru-GAL4* in female larvae, from which the CNSs were dissected to extract proteins for immunoprecipitation. The band representing Lola29F (arrow) was transferred to a PVDF membrane and subjected to Edman sequencing. **b** HPLC profiles of Edman sequencing of Lola29F. The 20 known amino acids served as standard (top-left panel). The first five amino acids of the N-terminal region of Lola29F were determined to be SSTAA, which corresponds to the a.a.264-a.a.268 in Lola29M. **c, d** Western blotting of S2 cell extracts probed with anti-Lola-exon 29 (upper panel). Administration of proteasome inhibitors MG-132 (**c**) or lactacystin (**d**) reduced the amount of Lola29F-like. α-Tubulin was used as a loading control (lower panel). Effects of inhibitors on the amounts of Lola29M and Lola29F-like are shown on the right-hand side. *$P < 0.05$ by the Kruskal–Wallis analysis of variance followed by Steel's nonparametric multiple comparisons. **e** Distribution of lysine residues throughout the Lola29M protein. **f** Western blot analysis of transfected S2 lysates for Lola29M and Lola29F-like revealed a marked decline in the amount of Lola29F-like with K41R, K44R, K47R and K67R substitutions. *$P < 0.05$ by the Kruskal–Wallis analysis of variance followed by Steel's nonparametric multiple comparisons. **g** Shown is a western blot of extracts obtained from the larval CNS probed with anti-V5 that recognizes the C-terminus of a Lola29M variant. Replacement of the lysine residue at a.a. 41 with arginine (K41R) in the Lola29M-encoding sequence blocked the production of Lola29F in females (right lane). **h** The hypothesis that Lola29F is derived from Lola29M by truncation of the N-terminus with a BTB domain. Edman degradation analysis identified the N-terminus of Lola29F-like as SSTAA, which corresponds to a.a. 264–268 of Lola29M. The four distal-most lysine residues (K41, K44, K47, and K67) are postulated to form a degron, a set of degradation signals. The box plot shows median and 10th, 25th, 75th, and 90th percentiles (**c, d, f**). Source data are provided as a Source Data file

*lola29m[Δ1-300]* encoding Lola29F-like in *fru-GAL4*-positive cells. We found that *lola exon 29-RNAi* effectively reduced the courtship index to ~40% of the value in control males (Fig. 4h), a level of inhibition comparable to that obtained with *lola-COM RNAi*, which knocked down nearly all types of *lola* mRNAs

(Fig. 1b). As mentioned above, our western blot analysis confirmed that both *lola-COM RNAi* and *lola-exon29 RNAi* effectively reduced Lola29M expression (Supplementary Fig. 6). Importantly, Lola29M[K41R] significantly ameliorated the suppressive effect of *lola-exon29 RNAi* on male courtship activities,

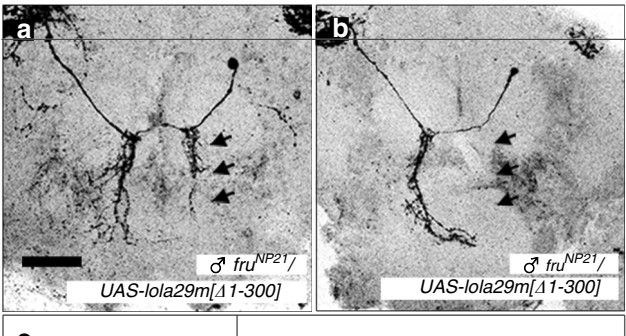

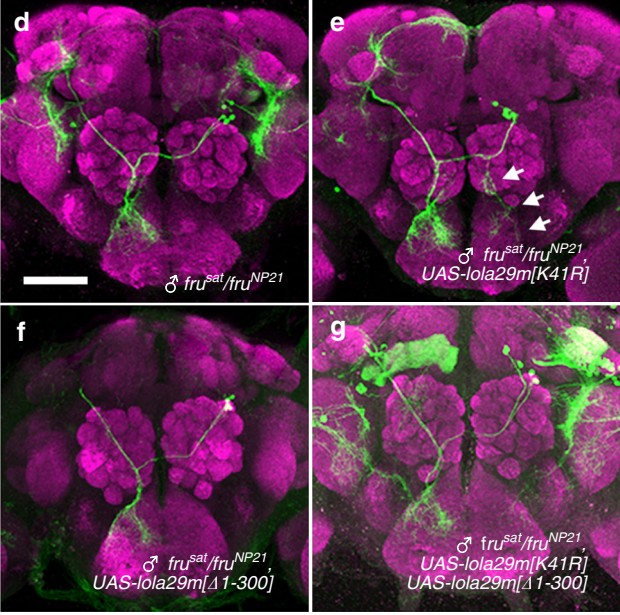

**Fig. 3** Lola29M promotes whereas Lola29F impedes male-typical neuritogenesis. **a–c** Analysis of single-cell mAL clones overexpressing Lola29F-like. Examples of single-cell clones (**a**, **b**) and the proportion of flies that carry single-cell mAL clones with or without the ipsilateral neurite (**c**). Scale bar: 50 μm. Statistical differences were evaluated by the Fisher's exact probability test (*$P < 0.05$). **d–h** mAL neurons in *fru* mutant (FruM protein-null) males without (**d**) or with overexpression of truncation-resistant Lola29M[K41R] (**e**) or Lola29F-like (**f**), or both (**g**). Representative MARCM clones are shown. **h** Quantitative comparisons of the percentage of mAL neuroblast clones that carried ipsilateral neurites **$P < 0.01$ by the Fisher's exact probability test. Scale bars: 50 μm

| Genotype | Single mAL neurons | |
|---|---|---|
| | With ipsilateral neurite (%) | Without ipsilateral neurite (%) |
| ♂ *fru^NP21^*/+ (n = 16) | 87.5 | 12.5 |
| ♂ *fru^NP21^/UAS-lola29m[Δ1-300]* (n = 12) | 50.0 | 50.0* |

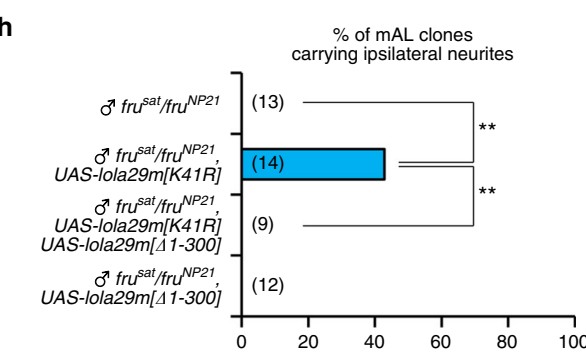

% of mAL clones carrying ipsilateral neurites

flies paired with test females of different genotypes were equally active in courtship (Fig. 4k), the observed reduction in mating success in this experiment was ascribable to a change in female receptivity. These results suggest that Lola29F and Lola29M play sex-specific functions in females and males, respectively, to ensure successful courtship and copulation. We note, however, that neither Lola29M nor Lola29F-like was able to resume the mating activity to the normal level in male or female flies expressing *lola-exon29* RNAi. The partial rescue of mating behavior by over-expressed Lola29M or Lola29F-like might suggest that mRNAs derived from the transgenes were also targeted by the *lola-exon29* RNAi to a certain extent.

**Lola29F prevents Lola29M from binding to its target**. To explore the mechanism by which Lola29F counteracts Lola29M, we first examined the possible effect of Lola29F/M on the transcription of *robo1*, a direct downstream target gene of FruBM[13]. Of note, Crowner et al.[18] have reported that a *robo1* mutant copy dominantly enhances the axon misrouting phenotype in a weak hypomorph of *lola*, *lola^ORE120^*, which barely manifests this phenotype on its own. Our quantitative RT-PCR experiment with male fly extracts revealed that overexpression of Lola29M decreased *robo1* transcripts, whereas overexpression of Lola29F-like had no effect on its own (Fig. 4l). Interestingly, the reduction in *robo1* transcript levels by Lola29M was not detected when Lola29F-like was co-overexpressed (Fig. 4l). In keeping with the RT-PCR result, overexpression of Lola29M[K41R] in the larval CNS reduced its immunoreactivity to the anti-Robo1 antibody (Supplementary Fig. 10). These results support the hypothesis that Lola29F inhibits the transcriptional repressor activity of Lola29M.

To further explore the mechanism by which Lola29F counteracts Lola29M, we conducted reporter assays. A series of reporters with a luciferase-coding sequence fused to *robo1* promoter fragments of varying length (Fig. 5a) were transfected into S2 cells with or without Lola29M and/or FruBM. The reporter activity was repressed by Lola29M, irrespective of whether FruBM was present (Fig. 5b), provided that the reporter contained a *robo1* promoter fragment of 0.9 kb or longer (Fig. 5c). Truncation-resistant Lola29M[K41R] was equally effective as wild-type Lola29M in repressing the *robo1* reporter activity (Fig. 5c). Notably, Lola29F-like encoded by *lola29m[Δ1-300]* had no effect on the *robo1* reporter activity (Fig. 5d). Intriguingly, the ability of Lola29M to repress reporter activities was completely blocked by the addition of Lola29F-like (Fig. 5e).

An *in silico* search for the putative Lola-binding motif within the Lola-responsive 0.9 kb region of the *robo1* promoter revealed an 18-bp direct repeat (DR), **G C A C T A A A** G A **G C A G G A A A**, which we named DR1 (Fig. 5f, g). The 0.9 kb reporter construct lost its sensitivity to Lola29M when DR1 was deleted from the promoter fragment (i.e., the 0.9 kb ΔDR1 reporter; Fig. 5h). Interestingly, DR1 was located immediately 3' to Pal 1, a palindrome sequence shown to be essential for a 42-bp

whereas Lola29F-like failed to do so (Fig. 4h). Conversely, in females, Lola29F-like overexpression via *fru-GAL4* mitigated the inhibitory effect of *lola-exon29* RNAi expression on female mating, whereas Lola29M[K41R] did not (Fig. 4i, j). Because male

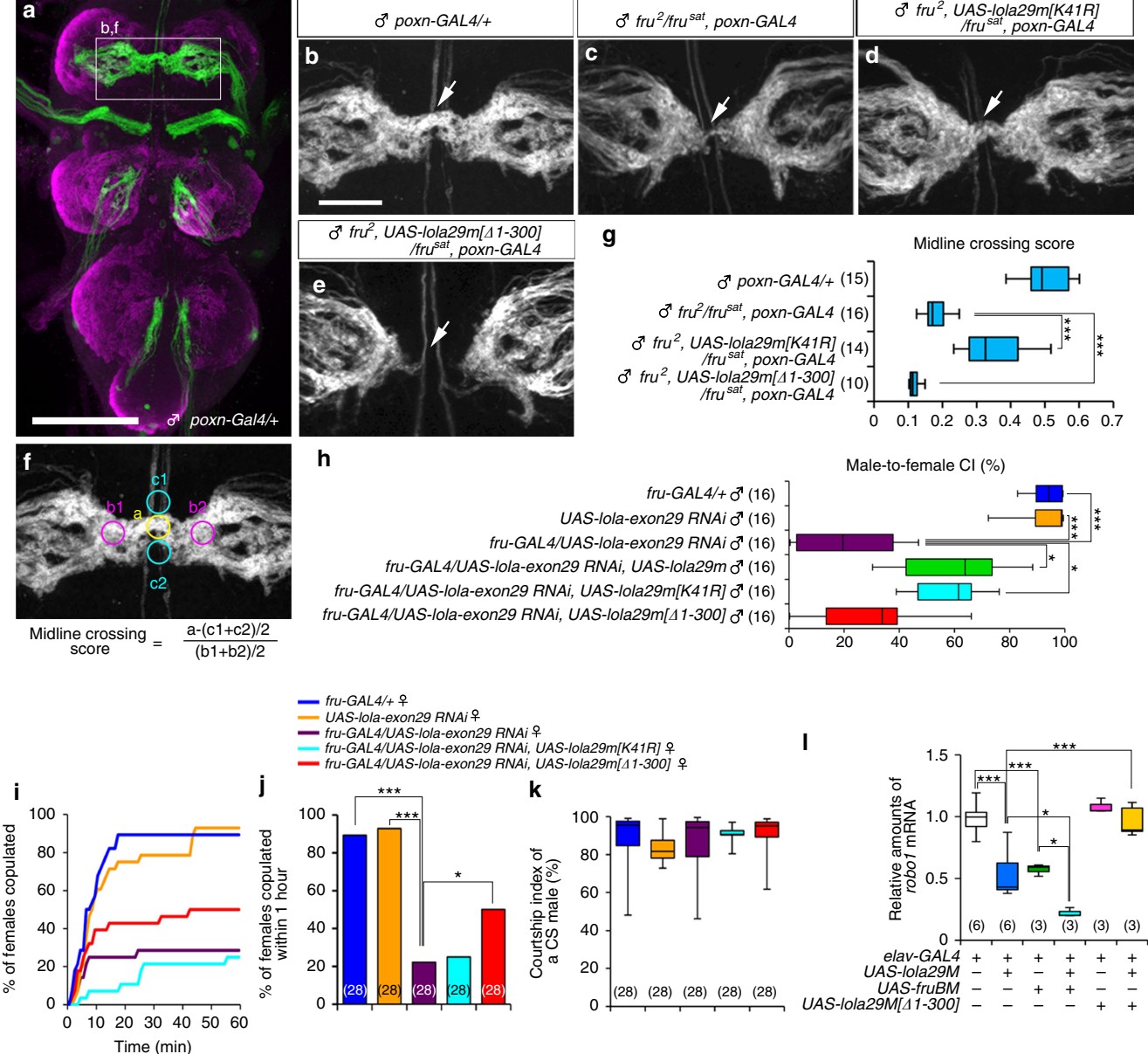

**Fig. 4** Lols29F/M are required for mating behavior. **a–g** Effect of Lola29M or Lola29F-like overexpression on midline crossing of leg sensory afferents. The whole thoracic ganglia are shown in (**a**), and the midline portion is enlarged in (**b–e**). The midline crossing score was calculated with the equation shown in (**f**) with values of fluorescent intensity measured at five regions, a, b1, b2, c1 and c2, near the midline. **g** Comparisons of the midline crossing scores among the indicated genotypes. The box plot shows median and 10th, 25th, 75th, and 90th percentiles. Numbers in parentheses indicate the number of scored flies. ***$p < 0.001$ by the Kruskal–Wallis analysis of variance followed by Steel–Dwass nonparametric multiple comparisons. Scale bars: 100 μm (**a**) or 30 μm (**b**). **h** Male courtship activities. *lola-exon 29* knockdown suppressed male courtship toward a female (3rd bar from the top). Overexpression of wild-type Lola29M (4th bar) or truncation-resistant Lola29M (5th bar) partially rescued the effect of Lola29M knockdown but Lola29F-like (bottom bar) had no rescuing effect. **i** Cumulative plot of the number of copulating females as a fraction of time after introducing a wild-type male and a female of the test genotype into an observation chamber. Knockdown by *lola-exon 29 RNAi* resulted in a reduced copulation success rate (~30%), which was partially rescued by overexpression of Lola29F-like (~50%) but not by overexpression of truncation-resistant Lola29M (~25%). **j** Maximal copulation success in a 1-h observation period. **k** Courtship indices for partner males used in this experiment. ***$P < 0.001$, *$P < 0.05$ by the Steel–Dwass post hoc test (**h**, **k**) and by the Fisher's exact probability test (**j**). **l** Effects of overexpression of 2 different *lola* transgenes and a *fruBM* transgene in neurons on the amount of *robo1* mRNA as measured by quantitative RT-PCR. Note that Lola29F-like inhibited the Lola29M action to reduce *robo1* mRNA. The numbers of examinations are indicated in parentheses. The box plot shows median and 10th, 25th, 75th, and 90th percentiles. ***$P < 0.001$, *$P < 0.05$ by the one-way ANOVA followed by Bonferroni's multiple comparisons (**l**)

FruBM-binding region, i.e., the FruBM response obligatory sequence (FROS)[13], to bind to FruBM (Fig. 5g). We conclude that DR1 is critical for the response of the *robo1* promoter to Lola29M.

To determine whether the observed reduction in *robo1* transcript levels by Lola29M overexpression is a result of

Lola29M binding to the *robo1* promoter, we carried out electromobility shift assays (EMSA) for *robo1* promoter DNA fragments with V5-tagged Lola29M[K41R] in the absence and presence of unlabeled competitor DNAs (Fig. 5i). We anticipated that Lola29M binds to the DNA fragment carrying FROS (probe DNA B in Fig. 5f). Our EMSA results supported this scenario. A

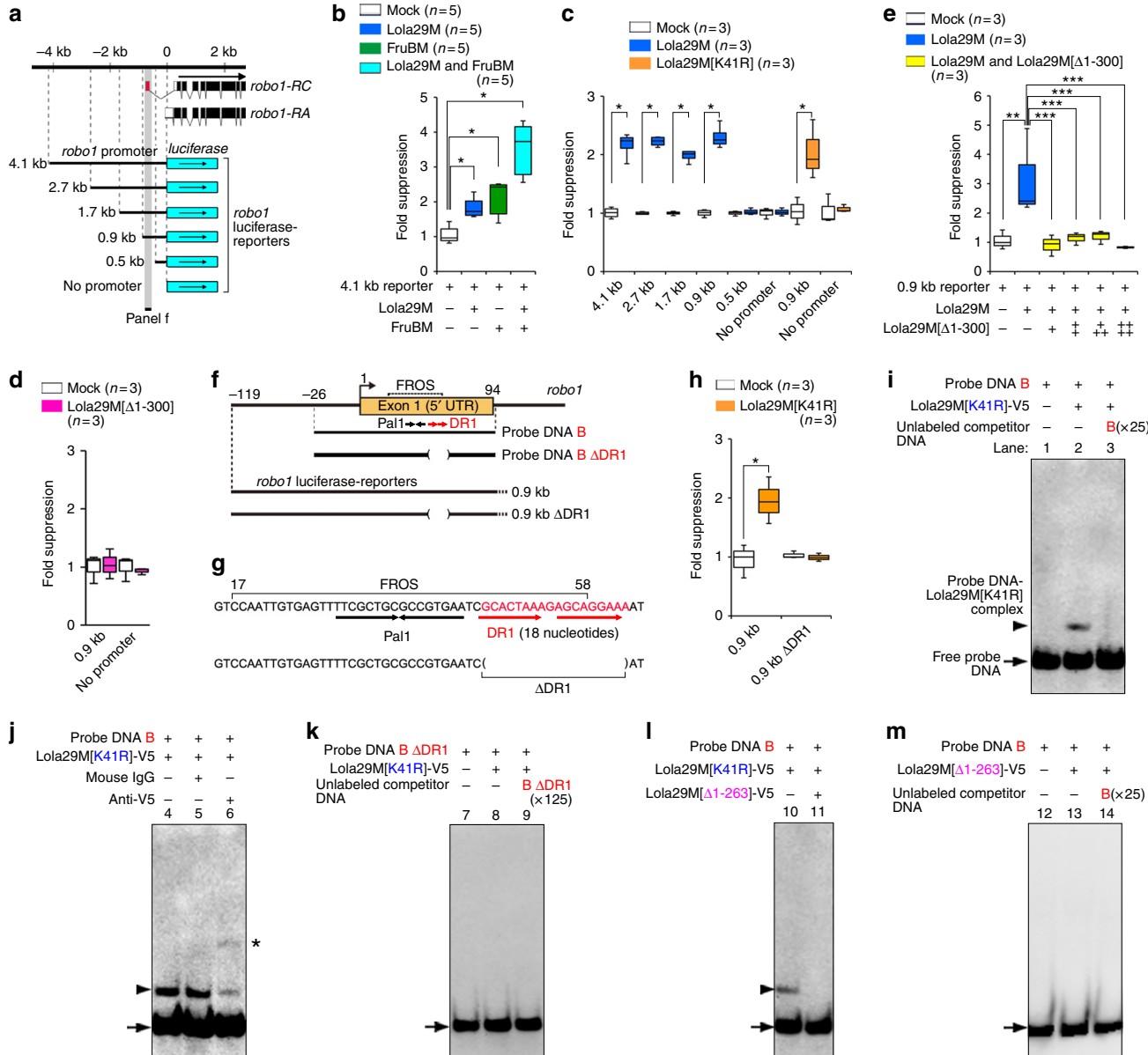

**Fig. 5** Lola29M/F molecular actions. **a–h** *robo1* promoter-luciferase reporter assays. **a** The *robo1* genomic map (upper) and reporter schematics (lower). The genomic distance (in kb) is indicated on the top with the reference point 0, at which the *robo1* genomic fragment was fused with the luciferase-coding sequence (blue box with an arrow). Below the thick line, the exon (box)-intron (thin line) organization of *robo1* is shown. Filled and open portions of the box represent the coding and non-coding regions, respectively. **b** The relative activities (ordinate) of a 4.1 kb *robo1*-promoter reporter cotransfected without (-) or with (+) Lola29M, FruBM or both. **c** The activities of reporters (ordinate) carrying a *robo1*-promoter fragment of different lengths (the fragment length is indicated on the abscissa) with (filled bars) or without (open bars) cotransfection of Lola29M or truncation-resistant Lola29M[K41R]. **d** The reporter activities were unaffected by cotransfection with Lola29F-like (Lola29M[Δ1-300]). **e** The suppression effect of Lola29M on reporter activities was inhibited by Lola29M[Δ1-300]. **f–m** Lola29M binding to the *robo1* promoter. **f** Genomic fragments used for EMSA. Probe DNA B and Probe DNA B ΔDR1 are shown in an expanded scale in the second and third rows. Reporters are schematically illustrated at the bottom. Pal 1: palindrome sequence, FROS: the FruBM-binding region, DR1: direct repeat 1. **g** The sequence around DR1 and FROS. The deletion in the 0.9 kb reporter is also indicated. **h** The repressor action of Lola29M[K41R] on the 0.9 kb reporter with or without DR1. *$P < 0.05$ by the Mann–Whitney $U$ test. **i–m** EMSA with fragment-B detected a retarded band (arrowhead) in the presence of truncation-resistant Lola29M[K41R] (lane 2), which was eliminated by adding the unlabeled fragment-B as competitors (lane 3). Mouse IgG had no effect (lane 5), whereas the Lola29M[K41R]-recognizing anti-V5 induced a super-shift (asterisk; lane 6). The ΔDR1 probe did not induce retardation (lanes 8). An arrow indicates free DNA. Lola29M failed to produce the retarded band in the presence of Lola29F (lane 11), which on its own, did not produce the retarded band either (lane 13). Source data are provided as a Source Data file

retarded band appeared when Lola29M[K41R] was present (Fig. 5i: lane 2 cf. lane 1), which was inhibited by the addition of unlabeled competitive DNA (Fig. 5i: lane 3). The mobility shift was not observed with mouse IgG unrelated to Lola (Fig. 5j: lane 5

cf. lane 4), whereas a super-shift was induced by the anti-V5 antibody that recognizes Lola29M[K41R] (Fig. 5j:* in lane 6). Notably, probe DNA B without DR1 (Probe DNA B ΔDR1) was unable to produce a retarded band (Fig. 5k). We conclude that

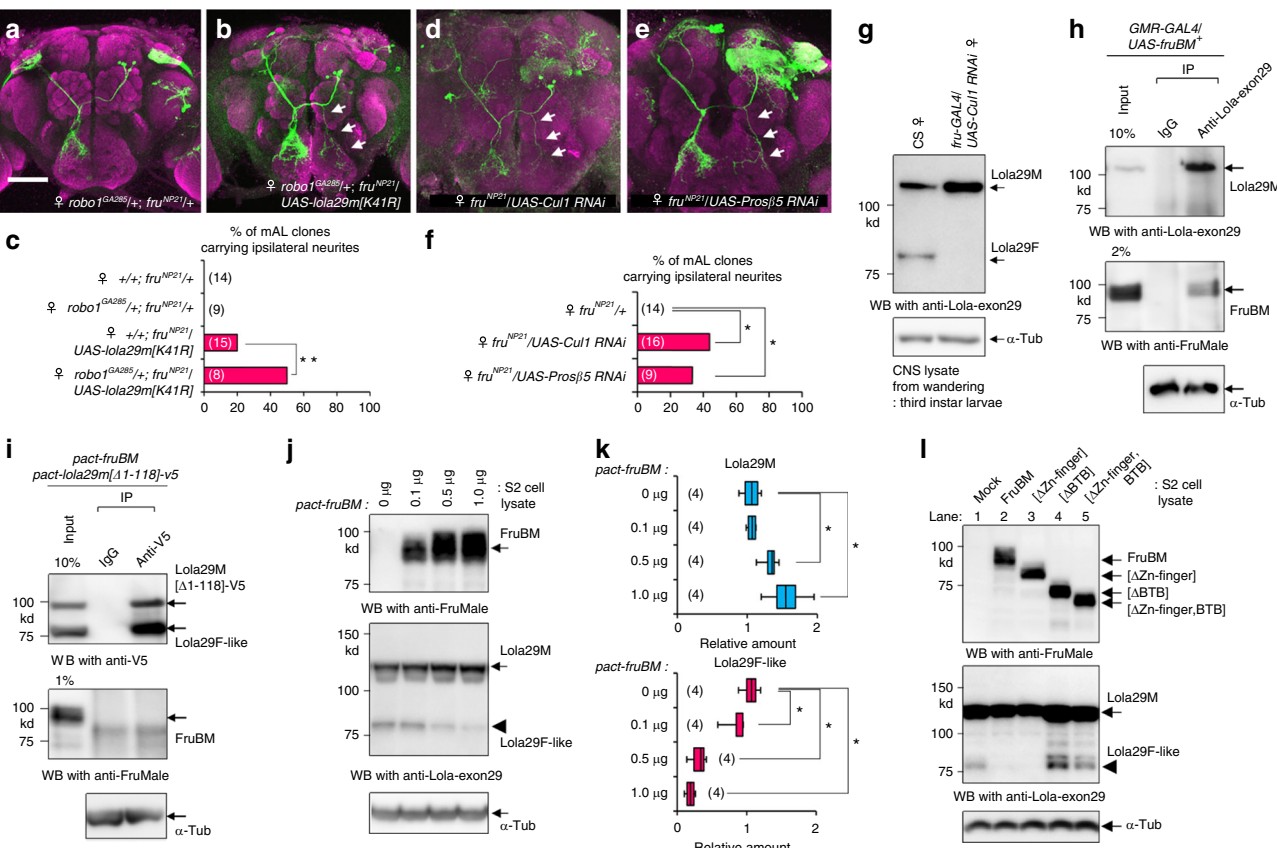

**Fig. 6** FruBM protects Lola29M from its ubiquitin-proteasome-mediated truncation. **a–c** mAL neuroblast clones in *robo1^GA285^/+* females without (**a**) or with (**b**) Lola29M[K41R] overexpression and quantitative comparisons of the number of clones carrying the male-specific neurite among the genotypes indicated (**c**). \*\**P* < 0.01 by the Fisher's exact probability test. **d–f** Knockdown of *Cul1* (**d**) or *Proteasome subunitβ5* (**e**) in females results in the formation of the ipsilateral neurite. Representative neuroblast clones with the ipsilateral neurite (**d**, **e**) that is otherwise absent from a female neuron. Scale bar: 50 μm. **f** Comparisons of the percentage of mAL neuroblast clones that carried ipsilateral neurites between females without and with *Cul1* or *Prosβ5* knockdown. \*ic*P* < 0.05 by the Fisher's exact probability test. **g** Western blotting with anti-Lola-exon 29 of larval CNS extracts with or without *Cul1* knockdown. **h–l** FruBM protects Lola29M from N-terminal truncation. **h** Overexpressed FruBM is coprecipitated with endogenous Lola29M. Anti-Lola-exon29 was used for immunoprecipitation and anti-FruMale for detection on western blotting. **i–l** Lola29M-FruBM interactions depend on the BTB domain of each. The antibody that recognizes the C-terminal V5 tag of Lola29M-V5 (anti-V5) precipitated N-terminally deleted Lola29M-V5 (Lola29M[Δ1-118]-V5) but not intact FruBM (**i**) in S2 lysates. **j** The abundance of Lola29F (middle panel) is inversely proportional to the amount of FruBM (upper panel) and also to that of Lola29M (middle panel) in S2 cells. The amount of *fruBM*-expressing plasmid (*pact-fruBM*) used for transfection is indicated above each lane. α-Tubulin (α-Tub) served as a loading control (lower panel). **k** Relative abundance of Lola29M (upper panel) and Lola29F-like (lower panel) in the presence of different amounts of FruBM (indicated at the left of the bars). \*P* < 0.05 by the Kruskal–Wallis analysis of variance followed by Steel's nonparametric multiple comparisons. **l** Deleting the Fru BTB domain resulted in Lola29F production (ΔBTB: lane 4; ΔZn-finger, BTB: lane 5), whereas deleting the Zn-finger motif (ΔZn-finger: lane 3) did not. Western blots of overexpressed proteins probed with anti-FruMale (upper panel), anti-Lola-exon 29 (middle panel) and anti-α-Tub (lower panel). The box plot shows median and 10th, 25th, 75th, and 90th percentiles

DR1 in the *robo1* promoter mediates Lola29M binding. These results collectively suggest that Lola29M directly binds to the *robo1* promoter to repress its transcriptional activity. However, Lola29M failed to produce the retarded band in the EMSA with the *robo1* promoter fragment in the presence of Lola29F (Lola29M [Δ1–263]-V5; Fig. 5l: lane 11 cf. lane 10), which, on its own, did not produce the retarded band either (Fig. 5m). We conclude that Lola29F inhibits Lola29M binding to the *robo1* promoter, and consequently derepresses *robo1* transcription. DR1 lies just outside FROS and, therefore, is dispensable for FruBM binding to the *robo1* promoter[13]. Remarkably, however, male flies homozygous for *robo1^Δ4^*, a *robo1* mutant carrying a 10 bp deletion that removes DR1, exhibited precocious wing switching during courtship (Supplementary Fig. 11 and Supplementary Movie 1), which represents a behavioral phenotype uniquely observed in flies with defects in the male-specific neurite formation of mAL neurons[13]. For example, *robo1^Δ1^* and *robo1^Δ2^*

mutations that delete a part of the core palindrome Pal1 in FROS dominantly induce the precocious wing switching in male flies[13]. The *robo1^Δ4^* mutation was recessive in inducing the precocious wing switching and had no dominant effect (Supplementary Fig. 11), suggesting that Lola29M bound to DR1 plays a distinct role that is needed for FruBM to fully repress *robo1* transcription.

Next, we examined whether a manipulation of the *robo1* gene dosage could modify the effect of Lola29M in vivo. More specifically, the effect of Lola29M[K41R] overexpression on the male-specific neurite formation was compared between females with the wild-type *robo1* and females heterozygous for a lethal loss-of-function allele, *robo1^GA285^*, which carries a premature termination codon in place of Q411 (http://flybase.org/reports/FBgn0005631.html). We found that the male-specific neurite was produced by overexpressed *lola29m[K41R]* at a higher rate in the presence of *robo1^GA285^* than in its absence (Fig. 6a, b), and the difference was statistically significant (Fig. 6c). Thus, the effect of

Lola29M[K41R] in promoting the male-specific neurite was enhanced by the *robo1* mutant heterozygosity, which, by itself, has no phenotypic effect. This finding supports the notion that *lola* and *robo1* interact in vivo.

**Lola29M cleavage involves the ubiquitin proteasome pathway**. To identify molecules involved in the proteolytic degradation of the Lola29 N-terminus, we analyzed the protein complex that was pulled down with Lola29M in immunoprecipitation assays. We overexpressed Lola29M that lacked a.a. 1–150 (Lola29M[Δ1-150]), as we thought that the N-terminal loss might stimulate further proteolysis of Lola29M. Mass spectrometric analysis identified 121 proteins that were precipitated with Lola29M, including Cul1, an E3 ubiquitin ligase[26], raising the possibility that this enzyme might be involved in the observed Lola29M truncation (Supplementary Table 1). Expression of Cul1 in mAL neurons was confirmed by the immunostaining with an anti-Cul1 antibody (Supplementary Fig. 12). We conducted a MARCM analysis in which *Cul1* was knocked down in mAL clones. Remarkably, mAL neurons with *Cul1* knockdown led to the male-specific neurite formation in females (Fig. 6d). Moreover, the production of Lola29F was markedly suppressed in females expressing *Cul1* RNAi via *fru-GAL4* (Fig. 6g). These observations are in keeping with the idea that Cul1 is involved in the N-terminal truncation of Lola29M.

Ubiquitinated proteins are known to be degraded by the 26S proteasome[27]. We thus examined the effect of knocking down *Proteasome subunit β5* (*Pros-β5*), a gene encoding a 26S proteasome subunit, on the mAL structure in the female brain. As shown in Fig. 6e, the expression of *Pros-β5* RNAi induced formation of the male-specific neurite in female mAL neurons (Fig. 6f). This observation is compatible with the hypothesis that the ubiquitin-proteasome pathway is involved in the N-terminal truncation and resulting conversion of Lola29M into Lola29F in female mAL neurons.

**FruM inhibits the Lola29M truncation and Lola29F formation**. A question arises as to why proteolysis of Lola29M occurs in females but not males. An interesting possibility is that the male-specific protein FruM protects Lola29M against proteolysis. First, we tested the possibility that Lola29M forms a complex with FruM, because both proteins have a BTB domain, through which they may interact directly with each other[28]. As shown in Fig. 6h, an anti-Lola-exon29 antibody precipitated FruM together with

Lola29M in flies. Moreover, a Lola29M-recognizing antibody failed to precipitate FruM, when the BTB domain of either FruM or Lola29M was deleted in an experiment with cotransfected S2 cells (Fig. 6i and Supplementary Fig. 13). These results support the hypothesis that Lola29M and FruBM form a complex through each other's BTB domains.

Second, we examined the effect of increased FruBM expression on the production of Lola29F in S2 cells transfected with *lola29m*. FruBM has been shown to be the major FruM isoform that contributes to neural sex differences[13,29]. We found that as the amount of FruBM increased the amount of Lola29F decreased (Fig. 6j, k). Notably, FruBM lost its ability to inhibit the Lola29F production when its BTB domain but not its zinc finger domain was deleted (Fig. 6l). These results demonstrate that FruBM binding to Lola29M, presumably as mediated by the BTB domains of two proteins, protects Lola29M against proteolysis, thereby preventing the production of Lola29F in males. Because female neurons lack FruM, Lola29M can be processed into Lola29F.

## Discussion

In this study, we showed that Lola29M, and its truncation product, Lola29F, play important roles in the masculinization and feminization of one of the three types of sexual dimorphism exhibited by *fru*-expressing mAL neurons: Lola29M promotes the male-specific ipsilateral neurite formation in males, whereas Lola29F counteracts Lola29M and prevents the male-specific neurite from forming in females (Fig. 7). Our results support the hypothesis that males have only the full-length protein Lola29M, because male-specific FruBM binds to Lola29M and protects it from truncation, whereas females have, in addition to full-length Lola29M, N-terminally truncated Lola29F due to the absence of FruBM. As a consequence, the Lola29M action is inhibited by Lola29F in females. This would mean that Lola29M exerts its masculinizing action only when FruBM is present, representing an efficient and secure means to induce masculine characteristics in *fru*-expressing neurons.

Our EMSA results demonstrated that Lola29M and FruBM share the same transcriptional target gene *robo1*, the transcription of which is repressed by each protein or by both proteins in synergy, according to our reporter assay data. The *robo1* gene encodes a transmembrane receptor of the immunoglobulin superfamily, Robo1[30], which is a key effector to prevent the male-specific ipsilateral neurite from forming in

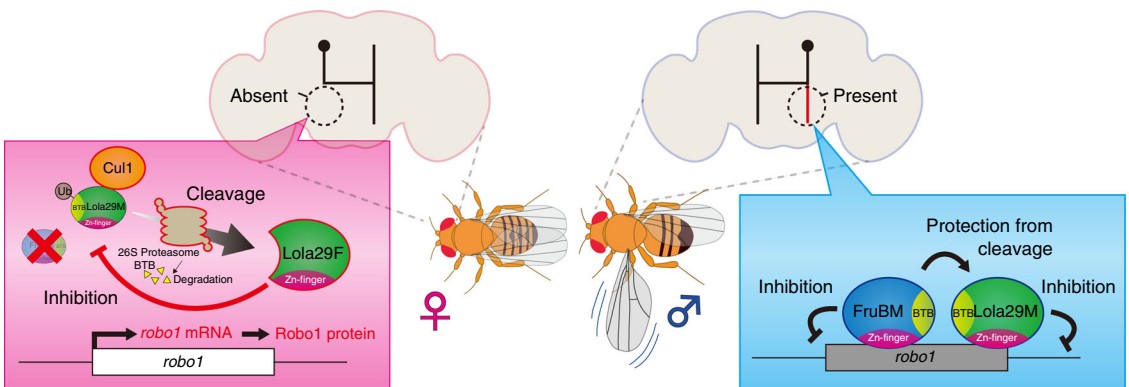

**Fig. 7** A model of the interaction between Lola29M and Fru in shaping the sexual dimorphism of mAL neurons and fly sexual behavior. Female neurons (left-hand side) lack FruBM, and consequently Lola29F is produced in sufficient abundance to impede the masculinizing action of Lola29M, leading to the production of mAL neurons without the ipsilateral neurite. Male neurons (right-hand side) express FruBM, which forms a complex with Lola29M and represses proteolysis of Lola29M. This results in the accumulation of Lola29M, which induces the ipsilateral neurite in some mAL neurons via *robo1* repression

females[13]. Lola29M induces the male-specific ipsilateral neurite by transcriptional repression of *robo1* (acting as a negative regulator of neuritogenesis) in males, whereas Lola29F appears to compete with Lola29M, allowing *robo1* to be transcribed, and as a result, Lola29F blocks the male-specific neurite formation in females. Our EMSA data identified two distinct DNA stretches, one harboring a Lola29M-binding site and the other a FruBM-binding site[13] in the *robo1* promoter region, which were located side-by-side. On the other hand, our immunoprecipitation assays indicated that Lola29M and FruBM form a complex in vivo. It might be that Lola29M and FruBM each bound to their own binding sites interact in *trans*, leading the *robo1* promoter to a conformation unfavorable for transcriptional activation.

We showed that the replacement of four lysine residues, K41, K44, K47 and K67, which are potential ubiquitination-targets, with arginine residues conferred resistance to N-terminal truncation on Lola29M, suggesting that the ubiquitin proteasome pathway is involved in the Lola29M processing. We also showed that the E3 ubiquitin ligase Cul1 is required for the N-terminal truncation of Lola29M, and that the male-specific neurite formation is induced by *Cul1* knockdown in female mAL neurons. In natural killer T-cell thymocytes, the E3 ubiquitination enzyme Cul3 is recruited to a chromatin-modifying complex, where it then induces changes in the ubiquitination patterns of other components of the same complex[26]. This recruitment of Cul3 to the chromatin-modifying complex is mediated by BTB zinc finger proteins such as PLZF and BCL6[26]. Therefore, it may be possible that Lola29M truncation is similarly mediated by a component in the protein complex to which Lola29M contributes. In this context, it would be worth mentioning that Cul1 and Cul3 play contrasting roles in the regulation of *hedgehog* (*hh*) signaling; in the absence of Hh, Cul1 is recruited to the Hh downstream zinc finger protein Cubitus interruptus (Ci155), which undergoes, as a consequence, partial degradation to yield the transcriptional repressor Ci75, whereas upon the Cul3 recruitment in the presence of Hh, Ci155 is completely degraded[31]. It remains to be determined how Cul1 is recruited to Lola29M for its partial degradation in *fru*-positive neurons.

A recent study documented neural activity-dependent activation of *lola* transcription, leading to the proposition that *lola* is an immediate early gene in insects[32]. The transient expression common to immediate early gene products is caused by the rapid degradation that follows their rapid induction. An intriguing possibility is that Lola proteins are vulnerable to rapid degradation through ubiquitination, unless their ubiquitination target sites are masked by a binding partner, such as FruBM, that interacts through the BTB domain. The known pleiotropic functions of Lola invite the supposition that other Lola isoforms expressed in a variety of cell types might form a complex with the different BTB proteins expressed there, accomplishing diversified roles unique to the individual cell type.

The present work thus provides new insights into the molecular mechanism whereby a common transcription factor such as Lola can play distinct roles in different cells in a context-dependent manner. A growing body of evidence indicates that fate changes in a variety of cell lineages in both vertebrates and invertebrates are induced by rapid epigenetic remodeling by chromatin regulators, such as BTB-zinc finger transcription factors, some of which are known to recruit E3 ubiquitin ligases to the chromatin-modifying complex and thus alter its ubiquitination pattern. It is therefore plausible that the proteolytic removal of the N-terminal BTB domain from these transcription factors within the chromatin-modifying complex is a prevalent means to convert the epigenetic marks.

Moreover, there is accumulating evidence that the association of BTB proteins with ubiquitin ligases plays multilayered regulatory roles in developmental decisions. For example, Germ cell-less (Gcl), a BTB protein conserved from *C. elegans* to humans with a transcriptional repressor activity[33,34], plays a key role in the soma-germ fate switch via a non-transcriptional mechanism. When complexed with Cul3, Gcl exits the nucleus to degrade the somatic fate determinant Torso and promotes the primordial germ cell fate in *Drosophila*[35]. In vertebrates, the Cul3-KBTB18 complex switches the translational program so as to specify the neural crest fate[36]. The present study further expanded the roles of the ubiquitin proteasome system—namely, the system was shown to function in the specification of sex-types of neurons via regulated proteolysis of a transcription factor that functions in the sex-determination molecular machinery.

## Methods

**Fly strains.** Flies were reared on cornmeal-yeast medium at 25℃. Canton-S served as a wild-type control. The following *fru* alleles were used: *fru*[sat], *fru*[2], and *fru*[NP21]. The GAL4 driver *fru*[NP21] used in this study is a recessive allele of *fru* induced by a P-element insertion into the *fru* second intron (Supplementary Fig. 1a), which exhibited no discernible defect in neurite structures. A subset of mAL neurons lacked the ipsilateral neurite in *fru*[NP21]/+ heterozygous males, as were the case in males of some flylight lines[37], which carried driver GAL4 insertions at genomic sites unrelated to the *fru* locus[3]. The GS2169 strain (#200307) carrying a *GS* vector inserted into the 5' region of the *lola* gene was obtained from the *Drosophila* Genetic Resource Center (Kyoto, Japan). *elav-Gal4*[C155] (*elav-Gal4*, #458), *robo1*[GA285] (#8755), *tra*[1] (#675), *UAS-Cul1-RNAi* (#29520), *UAS-Prosβ5 RNAi* (#34810) and other fly resources used in MARCM were obtained from the Bloomington *Drosophila* Stock Center. The *UAS-lola-COM RNAi* (#12573 and #12574) and *UAS-lola-exon29 RNAi* (#25335) lines were obtained from the Vienna *Drosophila* RNAi Center. The following fly lines were generous gifts from the indicated researchers: *lola*[ORE76] (an EMS-induced strong allele[18], Drs. M. L. Spletter and L. Luo); *lola*[03089] (a transposon-inserted hypomorphic allele[18], Dr. T. Aigaki); *insc-GAL4* (Dr. A. Knoblich); *fru-GAL4, 9-189-GAL4, UAS>stop>mCD8::GFP* and *fru*[FLP] (Dr. B. Dickson); *poxn-GAL4* (Dr. K.-i. Kimura).

**Modifier screens.** The female flies with both *GMR-GAL4* and *UAS-fru-typeB*[+] (denoted as *UAS-fruB*[+] in Supplementary Fig. 2) transgenes were crossed with male flies from *GS* P-element insertion stocks[15] or those from mutant stocks reported to have developmental defects in the nervous system (FlyBase: https://flybase.org/). In this screen, we overexpressed a FruCOM protein rather than a FruM protein when inducing the rough eye phenotype, as the former yielded more viable offspring. The nomenclature for Fru isoforms is adapted from our previous study[38] and different from that of other groups[39]. 1364 stocks were screened, resulting in 40 dominant suppressors of the rough-eye phenotype induced by *fru-typeB*[+] overexpression. Among the 5 Fru C-terminal variants, TypeB was most effective in rescuing the *fru*[sat] mutant phenotype[38] and thus was most likely to yield modifiers that were relevant to the in vivo functions of *fru*. Images of the compound eye surface were obtained with a scanning electron microscope (SU8000; Hitachi High-Technologies, Tokyo, Japan).

**Plasmids and transformants.** *lola29m* cDNAs with or without an amino acid replacement (point mutants include: *K41R, K44R, K47R, K67R, K74R, K77R, K105R, K113R, K127R, K138R, K149R, K181R, K255R, K256R,* and *K261R*; Fig. 2f) or with N-terminal deletions (deletants include: *Δ1-118, Δ1-150* and *Δ1-300*) were attached by the sequence for the N-terminal HA tag, and cloned into the *pMT/V5-His* C vector (Invitrogen). Subsequently, transgenic fly lines were generated with some of these constructs, i.e., *HA-lola29m[K41R]-V5* (Fig. 2g), *HA-lola29m[Δ1-300]-V5* (Fig. 1f) and *HA-lola29m-6xV5* (Fig. 2a), which were cloned into a *pJFRC81* vector[40] having a GFP coding sequence replaced by the respective *lola29m* cDNA sequence. The constructs thus generated were integrated into the *attP2* site of the *D. melanogaster* genome using commercial microinjection services (Best-Gene and Genetic Services). To construct *pact-FLAG-fruBM[ΔBTB]* and *pact-FLAG-fruBM[ΔZinc,BTB]* vectors (denoted as *ΔBTB* and *ΔZn-finger,BTB* in Fig. 6l), 111 amino acids (QQFCLRWNNHPTNLTGVLTSLLQREALCDVTLA-CEGETVKAHQTILSACSPYFETIFLQNQHPHPIIYLKDVRY-SEMRSLLDFMYKGEVNVGQSSLPMFLKTAESLQVRGL) composing the BTB domain were deleted from *pact-FLAG-fruBM*[9] or *pact-FLAG-fruBM[ΔZinc]*[13] respectively. To obtain *pact-lola29m-v5* and *pact-lola29m[Δ1-118]-v5* vectors (Figs. 5b and 6i), cDNA encoding *lola29m-v5* or *lola29m[Δ1-118]*-v5 was amplified from aforementioned *pMT* vectors and cloned into the *pact-MCS* vector[9].

**Antibody production.** A rabbit polyclonal anti-Lola-exon 29 antibody (Fig. 1c) was raised against a mixture of two 19-mer peptides, HARQEYIKIDTSRLEDKML and YRSDLRKHMNQKHADSGEA, which were both encoded by exon 29 of the

*lola* gene (residues 582-600 and 749-767 of Lola isoform Q (GenBank accession number AB107288)), and was affinity purified. A rabbit polyclonal anti-LolaCOM antibody (Supplementary Fig. 6) was raised against an 18-mer peptide, GGVAPKPESSGHHRGGKC, which was situated 4 amino acids distal to the BTB domain, within a region conserved in all isoforms of the Lola protein (residues 122-138 of the Lola proteins), and was affinity purified. The guinea pig polyclonal anti-FruMale antibody (Fig. 1i) was generated against a 19-mer peptide, HYAALDLQTPHKRNIETDV, in the male-specific N-terminus of Fru (residues 56-74 of FruM), and was affinity purified.

**Western blot assays.** For evaluating the specificity of the anti-Lola-exon 29 antibody (Supplementary Fig. 3), whole-cell extracts were prepared from *lola* heterozygous (*lola^{ORE76}/CyO-GFP*) or homozygous (*lola^{ORE76}/lola^{ORE76}*) embryos at developmental stages 11-14 by homogenizing them in a 4x sample buffer solution (Wako, 198-13282) containing 200 mM DTT. To determine the *lola*-knockdown effect by *UAS-lola-COM RNAi* or *UAS-lola-exon29 RNAi* (Supplementary Fig. 6), eye-antennal discs isolated from ten 3rd instar larvae were homogenized in the sample buffer. For the analysis of expression patterns of Lola proteins at the wandering 3rd instar larval stage (Fig. 1c), extracts were prepared by homogenizing the central nervous system from sexed individuals: males and females of Canton-S wild-type, males of *fru^{sat}fru^{NP21}* (selected as those not carrying a fluorescent balancer, *TM3 Sb Kr-Gal4 UAS-GFP*), females of *tra^1* homozygotes (selected as those not carrying a fluorescent Y–chromosome balancer, *Y-GFP*+[41], or males and females of the indicated genotypes (Fig. 1c) in the sample buffer. For Western blot analysis of S2 cell lysates (Fig. 2c, d, f), cells were disrupted by incubation with a lysis buffer (50 mM HEPES, pH 7.5, 300 mM NaCl, 50 μM ZnSO4, 10 mM NaF, 0.4% Nonidet P-40 (NP40) and cOmplete Protease Inhibitor (Roche)) for 1 h at 4 °C. Lysates were prepared by centrifugation and denatured in the sample buffer. After fractionation by SDS-PAGE and transfer to PVDF membranes (Life Technologies, IB401001), the blots were reacted with an antibody, rabbit anti-Lola-exon 29 (1:100; present study), mouse anti-V5 (1:500; Invitrogen, 46-0705), rat anti-HA (1:500; Roche, 11867423001), rabbit anti-FruMale (1:500)[38], rabbit anti-LolaCOM (1:200; present study), or mouse anti-α-Tubulin (1:500; Sigma, T6199), and subsequently by a horseradish peroxidase (HRP)-conjugated anti-rabbit, rat, or mouse IgG antibody (1:3,000; Sigma). Chemiluminescence was detected with Pierce Western Blotting Substrate Plus (Thermo Scientific, NCI32132) according to the manufacturer's instructions. Fluorescent images were obtained using ImageQuant LAS 4000 (Fujifilm). The signal intensity was quantified by using the software package ImageQuant TL (GE Healthcare). The endogenously expressed Fru proteins may affect transfection assays in S2 cells. We performed western blot analysis with S2 lysates for FruM expression several times, detecting no endogenous FruM expression (Supplementary Fig. 14). Even if Fru is expressed in S2 cells, the S2-derived Fru is likely non-sex-specific FruCOM rather than FruM, because the S2 cells originated from non-neural cells (probably lymphocytes), while FruM is specifically expressed in neural cells of males. We were unable to determine whether FruCOM is expressed in S2 cells due to the lack of an anti-FruCOM antibody that works in western blotting.

**Coimmunoprecipitation assays.** In the co-immunoprecipitation assays for Lola and Fru (Fig. 6h), we overexpressed *fru^+*-type BM in developing eye discs with the *GMR-Gal4* driver, as endogenous Fru proteins are expressed only in ~2% of neurons in the male nervous system and thus it is difficult to recover a sufficient quantity for analysis. CNS/eye-antennal disc complexes of wandering 3rd instar larvae were collected and homogenized in a cold lysis buffer (50 mM HEPES, pH 7.5, 300 mM NaCl, 50 μM ZnSO4, 10 mM NaF, 0.4% NP40 and cOmplete Protease Inhibitor (Roche)) for 1 h at 4 °C. Lysates were prepared by centrifugation and precleared with protein G beads (GE Healthcare, 17061801). After removal of the beads, the lysate was incubated with rabbit IgG (Sigma, I5006) or the anti-Lola-exon29 antibody in the aforementioned lysis buffer, which was modified to contain 150 mM NaCl / 0.2% NP40, for 3 h at 4 °C. The immuno-complexes were precipitated with protein G beads (GE Healthcare, 17061801) and analyzed by Western blotting with a primary antibody, *i.e.*, rabbit anti-FruMale (1:500)[38], rabbit anti-Lola-exon29 (1:100; present study), or mouse anti-α-Tubulin (1:500; Sigma T6199), and, as a secondary antibody, horseradish peroxidase (HRP)-conjugated anti-rabbit or mouse IgG (Sigma). In the assays shown in Fig. 6i and Supplementary Fig. 13, FruBM and Lola29M proteins with or without the indicated deletion (ΔBTB in FruBM, Δ1–118 in Lola29M) were co-overexpressed in S2 cells. 1 μg of each of the expression vectors was transfected into S2 cells (1x10[7] cells) using FuGene HD Transfection Reagent (Promega, E2311). Forty-eight hours after transfection, lysates were prepared by homogenizing in the aforementioned cold lysis buffer, then incubated with mouse IgG (Invitrogen, 10400C) or mouse anti-V5 antibody (Invitrogen, R960-25) in the lysis buffer with 150 mM NaCl and 0.2% NP40 for 3 h at 4 °C. The immuno-complexes were precipitated using protein G beads (GE Healthcare, 17061801) and analyzed by Western blotting with a primary antibody, i.e., mouse anti-V5 (1:500; Invitrogen, R960-25) or rabbit anti-FruMale (1:500)[38] and, as a secondary antibody, horseradish peroxidase (HRP)-conjugated anti-mouse or rabbit IgG (Sigma).

**Immunohistochemistry.** Immunostaining was carried out as described previously[9]. For immunostaining of the adult brain (Fig. 3d–g), the brains of 2- to 7-day-old flies of the relevant genotypes were dissected in phosphate-buffered saline (PBS), and fixed in 4% paraformaldehyde in PBS for 60 min on ice. After washing in PBT (PBS with 0.3% Triton X-100), the tissues were kept in PBTN (PBT with 10% normal goat serum) for 60 min on ice for blocking, and reacted with the primary antibody in PBTN for 24 h at 4 °C. After 2 h of washing in PBT, the tissues were incubated in the secondary antibody for 24 h at 4 °C. Samples were washed for 2 h in PBT before mounting with Vectashield (Vector Laboratories Inc.). To immunostain eye-antennal imaginal discs (Supplementary Fig. 2b) and the CNS (Supplementary Fig. 4 and Supplementary Fig. 10) of third instar larvae, the discs and CNS of relevant genotypes were dissected in PBS and fixed in 4% paraformaldehyde for 60 min on ice, and subsequent immunostaining was performed following the method described above. The antibodies used were as follows: rabbit polyclonal anti-GFP (1:500; Invitrogen A6455), mouse nc82 (1:10; Hybridoma Bank), guinea pig anti-FruMale (1:500; present study), rabbit anti-FruCOM (1:500)[38], mouse anti-Deadpan (1:100; Abcam, ab195173), mouse anti-Prospero (1:20, Hybridoma Bank), rabbit anti-Lola-exon29 (1:100; present study), mouse anti-Robo1 13C9 (1:50, Hybridoma Bank), chicken anti-GFP (1:500, abcam ab13970), rabbit anti-Cul1 (1:100, Invitrogen 71-8700) and secondary Alexa-488, 546, and 647 antibodies (1:500; Invitrogen). Stacks of optical sections at 1 μm were obtained with a Zeiss LSM 510 META confocal microscope and processed with NIH Image J (http://rsb.info.nih.gov/ij/) and Adobe Photoshop software.

**Clonal analysis of mAL neurons.** We used *fru^{NP21}-GAL4* to label mAL neurons (Fig. 1m–p). The somatic clones were produced using the MARCM method[42]. *y hs-flp / Y; FRT G13 UAS-mCD8-GFP / FRT G13 tub-Gal80; fru^{sat} / fru^{NP21}* was used for the *fru* mutant males (referred to as *fru^{sat} / fru^{NP21}* in Fig. 3d). The genotypes of flies used for the clonal expression of Lola29M[K41R] or Lola29M[Δ1-300] in *fru* mutant males (Fig. 3e–g) were as follows: *y hs-flp / Y; FRT G13 UAS-mCD8-GFP / FRT G13 tub-Gal80; fru^{sat} / fru^{NP21} UAS-HA-lola29m[K41R]-V5-6xHIS* or *fru^{NP21} UAS-HA-lola29m[Δ1-300]-V5-6xHIS*. For clonal expression of Lola29M[K41R] or Lola29M[Δ1-300] in *fru* heterozygous flies (Fig. 3a, b and Supplementary Fig. 9), flies of the following genotypes were used: *y hs-flp / w* (female) or *Y* (male); *FRT G13 UAS-mCD8-GFP / FRT G13 tub-Gal80; fru^{NP21} / +* (control referred to as *fru^{NP21} / +* in Supplementary Fig. 9a) or *UAS-HA-lola29m[K41R]-V5-6xHIS* (Lola29M[K41R] overexpression, Supplementary Fig. 9b) or *UAS-HA-lola29m[Δ1-300]-V5-6xHIS* (Lola29M[Δ1-300] overexpression, Supplementary Fig. 9c). The genotype of flies used in clonal Lola29 knockdown experiments (Fig. 1o, p) was: *y hs-flp / Y; FRT G13 UAS-mCD8-GFP / FRT G13 tub-Gal80; fru^{NP21} UAS-lola-exon29 RNAi* (VDRC#25335). The genotype of flies used in clonal co-expression of Lola29M and Lola29F-like (Fig. 3g) was: *y hs-flp / Y; FRT G13 UAS-mCD8-GFP / FRT G13 tub-Gal80; fru^{NP21} UAS-ha-lola29m[Δ1-300]-v5-his / fru^{sat} UAS-ha-lola29m[K41R]-v5-his*. The genotype of flies used in clonal Cul1 and Prosβ5 knockdown experiments (Fig. 6d, e) was: *y hs-flp / w; FRTG13 tub-Gal80 / FRTG13 UAS-mCD8-GFP; fru^{NP21} / UAS-Cul1 RNAi* (BL#29520) or *UAS-Prosβ5 RNAi* (BL#34810). The genotype of flies used in clonal expression of Lola29M[K41R] in *robo1* heterozygous females (Fig. 6a, b) was: *y hs-flp UAS-mCD8-GFP / w; FRT40A tub-Gal80 / robo1^{GA285} FRT40A; fru^{NP21} / +* (control, panel a) or *UAS-ha-lola29m[K41R]-v5-his* (Lola29M overexpression, panel b). For the production of neuroblast clones in mAL neurons, embryos at 0 to 24 h after egg laying (AEL) were heat shocked at 37 °C in a water bath for 1 h. For the production of single-cell clones of mAL neurons, larvae at 5–6 days AEL were heat shocked at 37°C for 1 h. Flies to be tested were reared at 29 °C after the heat shock in order to enhance the expression of transgenes. The number of clones examined was different from experiment to experiment as a consequence of the stochastic nature of mosaic generation. This might complicate interpretations of data, particularly in the experiment shown in Figs. 3c, 6c and 6f, where the proportion of neurons with the ipsilateral neurite was quantitatively compared. We therefore performed a post hoc power analysis of these data with the aid of Webpower[43], which yielded the following results: Power=0.7194, α=0.05 for Fig. 3c; power=0.3095, α=0.05 for Fig. 6c; power=0.9774, α=0.05 (*fru^{NP21}/+* vs. *fru^{NP21}/UAS-Cul1 RNAi*) and power = 0.8209, α = 0.05 (*fru^{NP21}/+* vs. *fru^{NP21}/UAS-Pros-β5 RNAi*) for Fig. 6. Here, the sample number is considered to be large enough when it gives a power value larger than 0.7. The data shown in Figs. 3c and 6f fulfilled this criterion but those in Fig. 6c did not. Our calculations indicate that power values over 0.7 would be obtained if 23 or more samples were subjected to analysis in the experiment shown in Fig. 6c (*cf.* 8-15, the current sample numbers). However, the flies with the genotype *robo1^{GA285}/+; fru^{NP21}/UAS-lola29M^{K41R}* were extremely difficult to obtain as most of the individuals died before eclosion. Because the statistical difference at *P<0.01* found in this set of data stands regardless of whether the power exceeds 0.7 or not, we consider that the result in Fig. 6c supports our conclusion that *robo1* and *lola* (Lola29M) interact in vivo.

**Midline crossing score analysis.** The MCS (Fig. 4g) was calculated as described previously[9]. Briefly, stacked images of each sample were summed up using ImageJ, and a circle of 8 μm (marked as "a" in Fig. 4f) was drawn on the resultant image so that the circle was centered at the prothoracic midline, where trans-midline axons are expected to run in males. The fluorescent intensity within the circle marked "a" was measured to quantify the level of midline crossing by fibers. Similarly, the fiber

tracts locating lateral to the midline were quantified within circles "b1" and "b2" for the fluorescent intensity. To normalize for the background fluorescent level, the areas with no fibers delineated by circles "c1" and "c2" were also measured. The MCS was calculated as: MCS = $[a - (c1 + c2) / 2] / [(b1 + b2) / 2]$.

**Behavioral assays**. The flies to be tested were reared individually in vials under a 12 : 12 h light:dark cycle. For analysis of male courtship behavior (Figs. 1a, b, 4h and Supplementary Fig. 11), virgin males of the indicated genotypes were collected at eclosion and aged for 5–7 days. Canton-S virgin females were similarly prepared as mating partners of test males. In the behavioral assays, a male of each genotype and a Canton-S virgin female were paired in a small chamber of 8 mm in diameter and 3 mm in height. The flies were video recorded for 5 min. The courtship index (CI) was determined as the percentage of time that the male courted the females during a 5 min observation period. In calculating the CI, the time spent for all courtship elements, i.e., orientation, tapping, following, wing extension/vibration and attempted copulation, was included. The wing switching index was estimated by the method as described in ref. [13]. For the analysis of female mating behavior (Fig. 4i–k), virgin females of each genotype were collected at eclosion and aged for 5–7 days. Each female fly was transferred to a small chamber (8 mm in diameter and 3 mm in height) with a Canton-S virgin male. The behavior of the fly pair was recorded using a video recorder. To estimate the level of female sexual receptivity, the cumulative number of copulating pairs in a 1 h observation period was counted and compared among the fly groups of different genotypes.

**S2 cell culture**. *Drosophila* S2 cells (Invitrogen, R690-07) were grown in Schneider's *Drosophila* medium (Gibco, 21720-024) supplemented with 10% fetal bovine serum (HyClone, SH30071.03, heat-inactivated) and 1% antibiotic–antimycotic solution (Gibco, 15240062). To determine the effect of proteasome inhibitors (Fig. 2c, d), approximately $2 \times 10^6$ cells were prepared in triplicate for transfection with a *lola29m* expression vector: the first set was treated with MG-132 (InvivoGen, tlrl-mg132, affinity purified and dissolved in DMSO) at concentrations of 0-50 μM for 5 hr, the second set was treated with Lactacystin (Kyowa Medex, OP18, dissolved in ethanol) at concentrations of 0-20 μM for 6 h and the third set was left without drug administration. 0–1 μg of *pact-fruBM* (Fig. 6j)[9], 1 μg of *pact-fruBM* with or without the indicated modifications (Fig. 6l), or 1 μg of modified *pMT-HA-lola29m-V5* (Fig. 2f) was transfected into approximately $1 \times 10^7$ cells per plate by using FuGENE HD Transfection Reagent (Promega, E2311). Forty-eight hours after transfection, the cells were disrupted by incubation with the aforementioned lysis buffer for 1 h at 4 °C and subjected to the analysis of expression profiles of Lola protein isoforms.

**RACE PCRs**. 5′ and 3′ RACE PCRs (Supplementary Fig. 5) were performed using the SMARTer RACE 5′/3′ kit (Clontech, Z4858N). Poly A(+) RNA was extracted from the CNS of a wandering third instar larva of a Canton-S female using the Micro-FastTrack 2.0 kit (Life Technologies, K1520-02). RACE-ready cDNA was synthesized from the extracted RNA. For 5′ RACE, the forward primer was a mixture of oligo named "5′ RACE primer (F)", which was used in conjunction with the reverse primer named "Primer 1 (R)", a gene-specific primer. For 3′ RACE, the forward primer was the gene-specific "Primer 2 (F), which was used in conjunction with the "3 RACE primer (R)", a mixture of oligo. For 3′ and 5′ RACE PCR, the following gene-specific primers were used: Primer 1 (accatccagcaatcgcagacgcatgc) and Primer 2 (cgcaggagcatcttgtcctccagcct), respectively. The cycling parameters for 3′ and 5′ RACE-PCR were as follows: 1 cycle of 94 °C for 2 min, 30 cycles of 94 °C for 30 seconds and 63 °C for 30 seconds and 68 °C for 2 min, followed by 68 °C for 10 min. DNA fragments obtained by 5′ and 3′ RACE PCR were subcloned into the *pGEM-T* vector and sequenced with ABI 3500 Genetic Analyzer (Applied Biosystems).

**Reporter assays**. Reporter assays employing the *robo1* promoter luciferase reporters (Fig. 5a–h) were carried out as described previously[13]. The *pGL3-promoter* vector carrying the 4.1, 2.7, 1.7, 0.9 kb (with or without 18 nucleotide deletion, ΔDR1), or 0.5 kb *robo1* promoter region[13], the *pRL-TK Renilla* luciferase vector (an internal control, Promega, E2241) and either the *pact-FLAG-fruBM* (a FruBM-expression vector)[9], *pact-FLAG-lola29M* (a Lola29M-expression vector; present study), *pact-FLAG-lola29M[K41R]* (a Lola29M[K41R]-expression vector; present study), *pact-FLAG-lola29M[Δ1-300]* (a Lola29M[Δ1-300]-expression vector; present study), or *pact-MCS* (an empty vector)[9] were co-transfected into S2 cell lines using FuGene HD Transfection Reagent (Promega, E2311). Forty-eight hours after transfection, the cells were lysed with a passive lysis buffer (Promega, E1910), and luciferase activity was measured using a Dual-Luciferase Reporter Assay System (Promega, E1910). To normalize the transfection efficiency, raw values for firefly luciferase activity were divided by raw *Renilla* luciferase activity values. The values representing fold suppression (x) were calculated by dividing the normalized luciferase-activity values (lucA) in cells cotransfected with the empty vector *pact-MCS* by those (lucB) in cells cotransfected with *pact-FLAG-fruBM* (x=lucA/lucB). All experiments were carried out in triplicate, and data are presented as the fold suppression in mean (±SEM) relative luciferase activity.

**qPCR**. qPCR (Fig. 4l) was performed using the LightCycler 1.0 system (Roche). Total RNA was extracted from the CNS of wandering third instar larvae using the RNeasy Mini Kit (Qiagen, 74104). To quantify *robo1* expression levels, equal amounts of RNA were used to synthesize cDNA using the ReverTra Ace qPCR RT kit (TOYOBO, FSQ-101). Each cDNA was mixed with SYBR Premix Ex Taq II (TAKARA, RR820S) and 5 pmol of both forward (5′-CCACGCTCAACTG-CAAAGTGGAG-3′) and reverse (5′-AACTGGACGCGGTGCGATTTCTT-3′) primers. *RpL32* (*rp49*) was amplified as an internal control using the primer pair: 5′-AGATCGTGAAGAAGCGCACCAAG-3′ (forward) and 5′-CACCAGGAACTTCTTGAATCCGG-3′ (reverse). qPCR was conducted at 95 °C for 30 sec (initial denaturation), followed by 40 cycles of denaturation at 95 °C for 5 s, annealing at 55 °C for 30 s and elongation at 72 °C for 30 s. Data processing was performed using LightCycler Software Ver. 3.5 (Roche).

**Edman sequencing**. To obtain a large amount of Lola29F sufficient for mass spectrometric analysis, we overexpressed the full-length *lola29m* tagged by V5 (*UAS-lola29m-6xV5*) under the control of *fru-GAL4* in transgenic *Drosophila*. The brain-ventral nerve cord (CNS) complex was dissected from, in total, 1,000 larvae, and protein extracts from these CNSs were prepared using RIPA buffer (50 mM Tris-HCl, pH8.0, 150 mM NaCl, 0.5% Na-deoxycholate, 1.0% Triton X-100, 0.1% SDS, 1 mM DTT, 1 μg/ml Leupeptin, 1 μg/ml Pepstatin, 0.5% Aprotinin), and subjected to immunoprecipitation as described above using 50 μl Dynabeads protein G (Invitrogen, 10003D) and 5 μg of the anti-V5 antibody (Invitrogen, 46-0705). The immunoprecipitate was then subjected to SDS-PAGE and transferred to PVDF membrane (APRO Science, BW-5201). After transfer, Lola29F visualized by coomassie staining (WAKO, 299-50101) was subjected to Edman degradation for protein sequencing. Protein sequencing was performed using Procise 494 HT Protein Sequencing System (Applied Biosystems, U.S.A.).

**Electrophoretic Mobility Shift Assays (EMSA)**. EMSA (Fig. 5i–m) was carried out as described previously[13]. The *pMT-3xflag-lola29m[K41R]* and *pMT-3xflag-lola29m[Δ1–263]* vectors were transfected together or individually into S2 cells, and the expression of proteins was induced by the addition of copper sulfate. The 3xFLAG-Lola29M[K41R] and 3xFLAG-Lola29M[Δ1–263] proteins were purified from S2 cell lysate using anti-FLAG resin columns in a FLAG HA Tandem Affinity Purification Kit (Sigma, TP0010) according to the manufacturer's protocol. EMSA experiments were performed using a DIG Gel Shift Kit (Roche, 03353591910). Binding reactions were established as follows: 1x binding buffer (20 mM Hepes, pH 7.5, 10 mM $(NH_4)_2SO_4$, 0.2% Tween20, 30 mM KCl, and 50 μM ZnSO_4), 0.1 μg/μl poly[d(I-C)], 5 ng/μl poly-L-lysine, 0.8 ng of a digoxigenin (DIG)-labeled probe containing the putative Lola29M-binding site, and 2 μl of purified 3xFLAG-Lola29M[K41R] and/or 3xFLAG-Lola29M[Δ1–263] protein in a final volume of 10 μl. For competition reactions, a 25-fold excess of unlabeled double-stranded oligonucleotide was included. Reactions were incubated at room temperature for 15 min. For antibody-binding assays, 1 μg of normal mouse IgG (Invitrogen, 10400C) or mouse anti-V5 (Invitrogen, 46-0705) was added to the binding reaction and the incubation proceeded for a further 15 min. DIG-labeled DNA-protein complexes were detected by chemiluminescence using an ImageQuant LAS 4000 system (Fujifilm).

**Mass spectrometric analysis**. Aiming at identifying molecules that contribute to the N-terminal truncation of Lola29M, we carried out mass spectrometric analysis of proteins precipitated with a partially truncated Lola29M in the presence of the anti-V5 antibody. More specifically, the *pMT-HA-lola29m[Δ1-150]-V5* vector was transfected into S2 cells, and the expression of proteins was induced by the addition of copper sulfate. The HA-Lola29M[Δ1-150]-V5 products were immuno-purified using Anti-V5-tag mAb-Magnetic Beads (MBL, M167-11), trypsinized, and then directly subjected to LC-MS/MS analysis as previously described[44]. Briefly, extracted peptides were analyzed by ESI-MS/MS using an LTQ velos Orbitrap ETD instrument (Thermo Fisher Scientific, Pittsburgh, PA, USA). MS spectra were recorded over a range of 321−1600 m/z, followed by data-dependent collision-induced dissociation (CID) MS/MS spectra generated from the 15 highest intensity precursor ions. For protein identification, spectra were processed using Proteome Discoverer 1.3.0.339 (Thermo Fisher Scientific) against MASCOT algorithm. For protein database searches, SwissProt (Taxonomy: Drosophila (fruit flies)) was used. The following parameters were used for the searches: tryptic cleavage, up to two missed cleavage sites, and tolerances of ± 10 ppm for precursor ions, and ± 0.8 Da for MS/MS fragment ions. MASCOT searches were performed allowing optional methionine oxidation and N-terminal acetylation, and fixed cysteine carbamido-methylation. Peptide data were filtered using a Mascot significance threshold of 0.05 as valid identification.

**Detection of ubiquitinated Lola29M**. To detect ubiquitination of Lola29M (Supplementary Fig. 8), Lola29M-3xFLAG was overexpressed in S2 cells. 1 μg of the *pMT-lola29m-3xFLAG* vector was transfected into S2 cells ($1 \times 10^7$ cells) with or without 1 μg or 3 μg of the *pMT-frubm* vector, and the expression of proteins was induced by the addition of copper sulfate. The Lola29M-3xFLAG protein was purified from S2 cell lysate using anti-FLAG resin columns in a FLAG HA Tandem Affinity Purification Kit (Sigma, TP0010). The immunoprecipitate was then

analyzed by western blotting with a primary antibody, mouse anti-Ub-K48 (1:500, abcam ab140601) and a secondary antibody, HRP-conjugated anti-mouse IgG (1:3000, Sigma). Fluorescent images were obtained using ImageQuant LAS 4000 (Fujifilm).

**Statistical analysis**. Statistical analyses were done by GraphPad Prism 7.0b software.

**Reporting Summary**. Further information on experimental design is available in the Nature Research Reporting Summary linked to this article.

## Data availability

The data sets generated for this manuscript are available from the corresponding author upon reasonable request. List of proteins that were immuno-purified with Lola29M protein is found in Supplementary Table 1. Primer sequence data sets are found in Supplementary Tables 2, 3. DNA probes used in EMSA experiments are included in Supplementary Table 4. The source data underlying Figs. 1, 2, 5 and 6 and Supplementary Figs 3, 5, 6, 7, 8, 13, 14 are provided as a Source Data file.

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

## Acknowledgements

We thank the Bloomington *Drosophila* Stock Center, *Drosophila* Genetic Resource Center at Kyoto, and Vienna *Drosophila* Stock Center for fly stocks, and Azusa Utsumi for secretarial assistance. This work was supported, in part, by Grants-in-Aid for Scientific Research from MEXT to D.Y. (Nos. 17K19371, 17H05935 and 16H06371), to K.S. (Nos. 17K07040, 25132702 and 24700309), to A.Y. (No. 16K08606) and to H.I. (No. 16K06985) and a Life Science Grant from the Takeda Science Foundation (D.Y. and K.S.).

## Author contributions

K.S. and D.Y. planned the project, K.S. designed and performed the experiments, A.Y. performed mass spectrometric analyses, K.S., G.T. and H.I. analyzed the data, and K.S. and D.Y. wrote the paper.

## Additional information

**Competing interests:** The authors declare no competing interests.

