## [Peer Review File · Nature Communications]

Reviewers' comments:

Reviewer #1 (Remarks to the Author):

Sato et al. present a study connecting well to previous work by the laboratory. Their conclusions are based on a number of well-established assays that they have developed over the years and published in several important papers.

In the present manuscript, they describe a novel and unexpected role for Fruitless, the transcription factor that determines male from female nervous system anatomy and behavior. The current belief in the field is that male- and female-specific isoforms of Fru would bind and regulate the expression of male- and female-specific genes, respectively. In contrast to this idea, Sato et al. show that male Fru directly binds to another transcription factor, Lola, to prevent its cleavage and thereby allowing it to repress the expression of Robo (the authors had previously shown that Fru binds the Robo enhancer in Ito et al., Current Biology). The main claims of the paper are as follows:

1. The transcription factor Lola, which itself comes as multiple isoforms, genetically interacts with Fru.
2. Male-specific Fru mutants express female isoforms of Lola instead of the male isoform.
3. Lola male and Lola female are products of sex-specific cleavage where Lola male gets processed by ubiquitin ligase Cullin and the proteasome into the female form.
4. Lola male and female isoforms are sufficient to determine male vs. female anatomy (some gender-specific neurites) and behavior.
5. The male isoform of Fru, here FruBM, directly binds and prevents that Lola male is cleaved into Lola female.
6. Male Fru and male Lola together repress the expression of Robo, which in turn allows the formation of male-specific neurites in the male brain and ultimately male-specific behaviors.

In addition, they determine a number of details such as the exact cleavage site of Lola, which further add to the quality of this paper.

Taken together, although a few small questions remain, this is an important and interesting study that merits the publication in Nature communications. I have only some minor comments:

- Please explain why FruBM is used rather than FruAM etc.
- In the abstract you start with naming Cullin etc., which highlights this particular gene. In my opinion, the emphasis should be on Lola more than on this finding.
- Does Lola female also compete with Fru or just Lola male?

Reviewer #2 (Remarks to the Author):

Analysis of the sexually dimorphic fru circuitry has proven to be an exceptionally powerful system for understanding how changes in circuit structures underlie changes in behavior. However, the mechanisms through which fru acts to specify dimorphic circuits remain poorly understood. Here Sato and colleagues describe a role for lola in the sex determination cascade, placing it downstream of fru, in the specification of appropriate male-specific projections. One lola isoform, lola29, is shown to encode two sexually dimorphic proteins, Lola29M (common to males and females) and a shorter Lola29F (found only in females). Edman sequencing and in vitro assays using proteasome inhibitors argue that these distinct proteins do not result from alternate splicing in males and females, as is the case in the generation of the sexually dimorphic forms of fruitless and doublesex, but instead result

from post translational cleavage directed by the E3 ubiquitin ligase, Cullin1. Surprisingly, in males Lola29M appears to be protected from degradation to the female form, not as a result of the transcriptional activity of male specific FruBM, but rather through direct binding of Lola29M to FruBM, an interaction that appears to prevent ubiquitination and cleavage. Using clonal analysis, the authors argue Lola29M is at least partially required for the formation of normal male-specific mAL neurites and sufficient to induce male-specific mAL neurites in females. Similarly, Lola29F can inhibit proper neurite formation in males. Finally, the authors demonstrate robo1, previously shown by this group to be an important target of FruBM, is also bound and inhibited by Lola29M, an activity directly antagonized by Lola29F. Together these observations extend upon the work from this group to understand how the fruitless regulatory cascade acts to generate sexually dimorphic circuitry. However, I have several important concerns regarding the clarity of the presented evidence and conclusions drawn from the data.

Major points

1. A number of the most critical experiments demonstrating the relationship between Lola29M, Lola29F, robo1, and Cullin1 in the specification of sexually dimorphic mAL neurons are performed using clonal analysis, in which a presumably stochastic subset of neurons is labeled. However, as this group has previously shown, mAL neurons are a heterogeneous population (Kimura et al, 2005), with only a subset of male neurons forming the ipsilateral neurite that is the basis of the assay used throughout this study. This heterogeneity poses a potential problem for clonal analysis since they are never examining the same neural populations—if the experiments were not performed to saturation so that all classes are sampled then problems in interpretation and statistical analysis arise. In a number of experiments, the number of clones analyzed differs across genotypes (e.g. 3c, 6c, 6f). How then were the numbers of clones chosen for analysis? The authors should perform a power analysis to determine the number of clones needed to get an accurate sampling of the mAL neuron subtypes and accordingly analyze a standardized number of clones for each experiment.

More troubling is the strengths of the mutant phenotypes. The analysis of dimorphic pheromone responsive neurons in the thoracic ganglion is proposed to offer independent support of the author's model in a distinct type of neuron, however the strength of the reported phenotypes is not particularly compelling. It seems strange that given the relatively low ratio of Lola29F to Lola29M found in wildtype females (Fig. 1c), that a sensitized (i.e. Fru hypomorph) background is required to visualize the effects of what is likely supra-physiological levels of Lola29F driven under Gal4 control. Similarly, the clonal analysis of mAL neurons only produced strongly impacted mAL morphology when Lola29M/F were manipulated in a Fru-mutant background (Fig. 3h). Taken together with the difficulties in interpreting how representative of the clonal analysis of mAL is, this raises concerns about the significance of Lola29's role in the establishment of dimorphic circuits.

2. The use of a variety to different experimental systems complicates the analysis. In particular the use of in vitro experiments, where in vivo experiments are equally feasible, make it difficult to understand whether or not the results will hold true in the fly.

A specific point of confusion is that S2 cells are male cells that, according to modEncode data, express Fru (Flybase). Why then does S2-cell lysate produce Lola29F in absence of proteasome inhibitors? These results either confound the model or reflect a peculiarity of S2 cells that would require them to be validated for their suitability in this study.

Furthermore, while the EMSA assays testing Lola29M/F robo1 binding and inhibitory function offer nice in vitro evidence, significantly stronger support could be provided with in vivo CHIP-qPCR using the same animals assayed for robo1 expression performed in Fig. 4I.

3. Some of the described experiments are vague regarding the methodologies used and their outcomes. For instance, the screen used to identify *lola* lacks a clear description of other positive hits (e.g. was *robo1* found to be a suppressor of the *GMR>FruBM* eye phenotype?) and other genes that were identified. The mass spec analysis of the N-terminal *Lola29M* co-IP is similarly lacking in detail. Of particular importance, how was *Cullin1* selected for further analysis, given that it was ranked 31 of the 121 interacting proteins identified, most of which appear to be housekeeping genes?

Minor points

4. It should be demonstrated, if possible, that *Cullin1* and *Lola29F* are expressed in *mAL*

5. Since *lola* is broadly expressed, a nice demonstration of their model would be that *Lola29F* is present in *Fru*- neurons.

6. In the introduction the authors state: "Here we demonstrate that the male-biased *Lola29M* isoform forms a complex with *FruBM* to repress transcription from *robo1*...", which suggests that they believe that the *FruBM*-*Lola29M* interaction is necessary for both proteins' inhibitory roles and not merely for the protection of *Lola29M* from degradation. The model in figure 7 further supports this impression. This conclusion is unsupported by the evidence and the two proteins may both inhibit transcription independently from one another (supported by the observation that *Lola29MK41R* promotes ipsilateral neurite formation in females, independent of any *FruBM* product). The authors should clarify their model in the text and figure 7 to eliminate any potential confusion.

7. It would be nice to see that *Lola29MK41R* overexpressing clones show reduced *robo1* staining compared their non-clonal neighbors in females and that the reciprocal is true in *Lola29F* overexpressing clones in males.

Reviewer #3 (Remarks to the Author):

Fruitless (*Fru*) encodes sexually dimorphic transcriptional factors. *Fru* isoforms that occur exclusively in male (i.e., *FruMs*, *FruBM*) are shown responsible for male-specific neural development as well as behaviors. To understand how *FruM* functions in molecular levels, Sato et al. performed a screen to search genetic modifier of the *FruBM*-overexpression phenotype. This screen identified *longitudinals* lacking (*lola*), a gene encodes transcriptional factors implicated previously to regulate the neural development. *Lola* seems expressed in differentiating neurons including those expressing *FruM*. *Lola* gene produces many isoforms through RNA splicing, and *Lola* isoforms derived from a mRNA containing exon 29 (*Lola29*) were analyzed in the high resolution. Knocking down entire *Lola* isoforms or *Lola29s* in *fru-Gal4* neurons significantly attenuated sexual motivation of males measured by courtship index (CI) and concomitantly reduced male-specific neurites in *mAL* neurons, a subset of *Fru* neurons with sexually dimorphic neurites.

Lola29 seems to exist in two forms, *Lola29M* and *Lola29F*. *Lola29F* occurs exclusively in females and is a product of ubiquitination-mediated degradation of N-terminus (a.a. 1 – a.a. 263) of *LolaM*. *LolaM* occurs in both sexes but is more enriched in male than in the female. In the absence of functional *fru* alleles (i.e., in females or *fru* mutant males), some *Lola29M* is degraded to produce *Lola29F*. In males, *FruBM* seems to protect *Lola29M* from the degradation to prevent the production of *Lola29F*. Intriguingly, *Lola29M* and *Lola29F* are important for male courtship behavior and female mating receptivity, respectively, but this is not true in the other way around. *Lola29F* seems to block the developmental and behavioral function of *Lola29M*. Previously, *FruBM* was shown to bind a promoter region of *robo1* and result in suppression of *robo1* transcription, which produces male-specific neurite

in mAL. Likewise, Lola29M also binds a promoter region of robo1 via an 18bps-long direct repeat (DR1) that occurs immediately next to 3'end of the FruBM binding site, Pal1. Lola29M but not Lola29F can bind with FruBM through N-terminal BTB domains both share. Lola29F, an N-terminal truncated form of Lola29M does not have BTB domain. Together, authors proposed a model that the FruBM and Lola29M form an epigenetic complex that occupies a promoter region of robo1 to suppress robo1 transcription and as a consequence promote the formation of male-specific neurites in mAL and other fru neurons.

Authors made a convincing explanation on how a male-specific Fru isoform interacts with another transcriptional factor Lola29M to produce the male-specific characteristic of mAL neurites. The experiments were designed well and performed elegantly. The quality of data is outstanding. The results support the conclusions sufficiently. Narrations and figures are easy to understand.

1. Why did authors focus on Lola isoforms with exon29 among many others? Authors need to provide a rationale. Otherwise, authors may need to examine at least a few other Lola subtypes to evaluate sub-type specific role of Lola29M/F.
2. Does Lola29M affect the stability of FruBM? Did authors evaluate the possibility that Lola29M may also increase the activity of FruBM by protecting FruM from degradation? Is it possible that knockdown of Lola29M reduce FruBM levels?
3. Can other FruM-types (type A, type E) also interact with Lola29M, at least in protein levels?
4. Fig. 3c: Why lola29m[D1-300] cannot inhibit production of ipsilateral neurite completely? Is this because FruMs override Lola29F function or FruMs function independently (at least partially) from Lola29M? This result needs an explanation.
5. How does Lola29F (i.e., Lola29M[D1-300]) block binding of Lola29M on robo1 promoter? Does inhibitory function of Lola29F require Zn-finger domain? Can Lola29F expression alone compromise the male courtship vigor as well?
6. Fig 4h, 4i: Overexpression of lola29m[K41R] or lola29m[D1-300] did not fully restore male courtship vigor or female mating receptivity either. Is this because lola-exon29-RNAi also reduces expression of exogenous lola29m[K41R] or lola29m[D1-300]. If so, how do authors rule out the possibility that lola-exon29-RNAi suppress exogenous lola29m[K41R] and lola29m[D1-300] in different extents?
7. Line 315: How does 26S proteasome degrade Lola29M only partially but yet precisely? Does 26S proteasome have some sequence specificity? Is there any known example for such specific degradation? If so, the discussion should include such case.
8. Does the loss of DR1 affect the binding of FruBM on FROS site? Previous work from the authors' group (Ref# 19) may answer this question. Authors need to clarify this point in the discussion.

Minor comments

Line 25-27: Authors need to elaborate this point in the discussion or the introduction

Line 38: How do authors produce Supplemental Fig.1? Where does the data (about splicing isoforms) of lola come from? Please clarify the sources.

Line 76. ... well established: Add reference.

Line 87. ... identified to date: Add reference.

Line 91-92. The isoform ... as type-Q: I cannot find this information in Ref14. Please clarify this.

Ref14 reported the in vivo evidence indicating that lola interacts with robo. Authors need to introduce or discuss the relevant previous observations with sufficient details.

Line 96: Where does Lola29F occur in the female brain? Does it occur in fru neurons? Need to show the actual data.

Line 124: how do authors predict the truncation position between residues 250 and 300?

Lines 125, 158: No mass spectrometry data here.

Line 173: Need to show that Lola29M is indeed ubiquitinated.

Line 209. "female counterparts terminate in the contralateral neuromere": The male fibers seem to terminate in the contralateral neuromere, but female ones do not.

Line 304: difficulty in following this logic. Is this because Cullin1 cannot access the full-length Lola29M bound with FruBM?

Line 396-400: Elaborate this point with more specific examples.

Line 875: NS is absent in Fig. 2.

Line 955: P1 and P2 are absent in Fig 5.

Line 958: It should read as DeltaDR1. '1' is missing.

Fig. 1d: There is virtually no signal of Lola29F in CS. How could authors normalize data with zero value of CS?

Fig. 1l: It is not clear which cells are positive for GFP in Fig. 1l. Please label them.

Fig. 2h. What does the shield mark indicate? Specify it.

Fig 4l: Which sex was used to generate this figure?

Fig 5b, c, e, d, h: Fold suppression is difficult to comprehend. Provide formula how to produce it either in the figure legend or Materials and Methods.

Supplemental data

Line 44. "... right-hand side) in (c-g);" There is no scale bar in Supplemental Fig2 d, e, f, g.

Supplemental Fig. 4: need to include double labeling of anti-Lola29 and anti-Pros.

Supplemental Fig. 5c: What 5'RACE primer was used.

Supplemental Fig 5d: Show male and female data separately.

Line 93-94. No need to use capital letters.

Reviewer 1.

Q1. Please explain why FruBM is used rather than FruAM etc.

A1. FruBM is the most prevalent of the FruM isoforms and the requirement of FruBM in mAL neurons for the male-specific neurite formation has been well demonstrated. For these reasons, we focused on FruBM in this study.

Q2. In the abstract you start with naming Cullin etc., which highlights this particular gene. In my opinion, the emphasis should be on Lola more than on this finding.

A2. We modified the abstract so that it faithfully reflects the story line of the main text. Consequently, less emphasis is given to the involvement of Cullin1 in the processing of Lola.

Q3. Does Lola female also compete with Fru or just Lola male?

A3. We did not carry out an experiment to examine whether Lola29F competes with Fru, because Lola29F should not be produced in the presence of Fru, and therefore, it is unlikely that Lola29F and Fru coexist under normal conditions.

Reviewer #2 (Remarks to the Author):

Q4. A number of the most critical experiments demonstrating the relationship between Lola29M, Lola29F, robo1, and Cullin1 in the specification of sexually dimorphic mAL neurons are performed using clonal analysis, in which a presumably stochastic subset of neurons is labeled. However, as this group has previously shown, mAL neurons are a heterogeneous population (Kimura et al, 2005), with only a subset of male neurons forming the ipsilateral neurite that is the basis of the assay used throughout this study. This heterogeneity poses a potential problem for clonal analysis since they are never examining the same neural populations; if the experiments were not performed to saturation so that all classes are sampled then problems in interpretation and statistical analysis arise. In a number of experiments, the number of clones analyzed differs across genotypes (e.g. 3c, 6c, 6f). How then were the numbers of clones chosen for analysis? The authors should perform a power analysis to determine the number of clones needed to get an accurate sampling of the mAL neuron subtypes and accordingly analyze a standardized number of clones for each experiment.

A4. We are not entirely convinced by the reviewer's suggestion that the heterogeneity in mAL neurons could lead to a false positive judgment of statistically significant differences in the proportion of neurons with the ipsilateral neurite (male-specific neurite) for the following reasons. Among the three

data sets (Figures 3c, 6c and 6f) referred to by the reviewer, two (Figures 6c and 6f) were obtained from female flies, in which none of the mAL neurons carry the ipsilateral neurite unless they are genetically manipulated. In other words, no heterogeneity exists in the normal mAL neurite pattern in these cases, and therefore, the reviewer's argument does not stand here. The remaining data set (Figure 3c) was obtained from male flies, in which 12.5% of mAL neuron clones have no ipsilateral neurite (n=16); this agreed with our previous finding that 17.6% of mAL neuron clones were without the ipsilateral neurite (n=17; Ito et al., 2016) when a heat-shock regimen for inducing recombination similar to the one used in the present work was adopted (heat-shock at 37°C for 1 hr at 3-6 days after egg laying). We detected a significant increase in the incidence of obtaining single cell mAL clones that lack the ipsilateral neurite in the experimental group, i.e., a 50% increase (n=12). In this experiment, the level of statistical significance was $P < 0.05$ by the Fisher's exact probability test. A larger sample size would provide a more robust statistical difference, rather than a reduced level of statistical difference. On the other hand, a power analysis has typically been used in cases where a statistical test indicates no statistical difference between data sets with relatively small sample sizes. In such cases, the experimenter would like to know the likelihood of the statistically significant difference remaining if a larger number of samples was subjected to analysis. Because the differences of values in Figs. 3c, 6c and 6f were all statistically significant, we feel there is no convincing rationale for conducting the power analysis. We have thus decided to retain the interpretation and statistical analysis from the original submission. Nonetheless, we performed a post hoc power analysis of these data with the aid of Webpower (Zhang & Yuan, 2018, Practical statistical power analysis using Webpower and R, ISDSA), which yielded the following results.

Figure 3c:

Power=0.7194, alpha=0.05

Figure 6c:

Power=0.3095, alpha=0.05

Figure 6f:

fru[NP21]/+ vs. fru[NP21]/UAS-Cul1 RNAi: Power=0.9774, alpha=0.05

fru[NP21]/+ vs. fru[NP21]/UAS-Pros-beta5 RNAi: Power=0.8209, alpha=0.05

The sample number is considered to be large enough when it gives a power value larger than 0.7. The data shown in Figs. 3c and 6f fulfill this criterion but those in Fig. 6c do not. Our calculations indicate that power values over 0.7 would be obtained if 23 or more samples were subjected to analysis in the experiment shown in Fig. 6c (cf. 8-15, the current sample numbers). However, the flies with the genotype robo1[GA285]/+; fru[NP21]/UAS-lola29M[K41R] were extremely difficult to obtain as

most of the individuals died before eclosion. As discussed above, the statistical difference at $P < 0.01$ found in this set of data stands regardless of whether the power exceeds 0.7 or not, and thus we consider that the result in Fig. 6c supports our conclusion that robo1 and lola (Lola29M) interact in vivo.

Q5. More troubling is the strengths of the mutant phenotypes. The analysis of dimorphic pheromone responsive neurons in the thoracic ganglion is proposed to offer independent support of the author's model in a distinct type of neuron, however the strength of the reported phenotypes is not particularly compelling. It seems strange that given the relatively low ratio of Lola29F to Lola29M found in wildtype females (Fig. 1c), that a sensitized (i.e. Fru hypomorph) background is required to visualize the effects of what is likely supra-physiological levels of Lola29F driven under Gal4 control. Similarly, the clonal analysis of mAL neurons only produced strongly impacted mAL morphology when Lola29M/F were manipulated in a Fru-mutant background (Fig. 3h). Taken together with the difficulties in interpreting how representative of the clonal analysis of mAL is, this raises concerns about the significance of Lola29's role in the establishment of dimorphic circuits.

A5. Although the reviewer considered that “the strength of the phenotypes is not particularly compelling”, our statistical analysis indicated a very significant difference in the midline crossing score between the control and experimental groups at the significance level of $P < 0.001$. We showed that Lola29M overexpression can induce the ipsilateral neurite only in the sensitized genetic background, i.e., fru[NP21]/fru[2], and the reviewer considered this another “evidence” that the Lola29M has only a weak effect (because it could not show its effect in the fru-wild-type background [fru+]). Please note that the masculinizer effect of overexpressed Lola29M cannot be demonstrated in fru[+] males, because these males develop a full-sized ipsilateral neurite even without any manipulation, and thus there is no room for further masculinization. Although we did not include the data for the masculinizing experiment in the fru-null mutant background, we conducted an experiment in females, the neurons of which completely lack Fru expression. As expected, Lola29M overexpression in females induced the ipsilateral neurite as shown in Supplementary Figure 9. In the experiment shown in Figure 3h, which caught the reviewer's attention, we chose the fru null background so that any changes in an opposite direction (masculinization vs. feminization) would not be overlooked. Indeed, did we detect masculinization upon Lola29M overexpression and the suppression of this masculinization effect upon an additional Lola29F overexpression. In the presence of functional FruM (e.g., fru [+] males), endogenous Lola29M will be protected from degradation by

bound FruM, allowing it to repress transcription from the target gene *robo1*, with a concomitant formation of the ipsilateral neurite (note that *robo1* inhibits the ipsilateral neurite formation, which is relieved from inhibition by FruM-mediated transcriptional repression of *robo1*). The reviewer here again expressed her/his concern about the difficulties in interpreting clonal data. In our opinion, the interpretation of the MARCM data shown in Figure 3h and Supplementary Figure 9 is straightforward, because both experiments were designed to compare the test group subjected to the masculinization treatment with the control group in the “default” phenotype state, i.e., the complete lack of the ipsilateral neurite. For this reason, we used neuroblast clones in which all cells composing an mAL cluster are stained (thus raising no probability issue at all), unlike in the cases with single cell clones. Based on these considerations, we have decided to keep the relevant passages and figures from the original submission.

Q6. The use of a variety to different experimental systems complicates the analysis. In particular the use of *in vitro* experiments, where *in vivo* experiments are equally feasible, make it difficult to understand whether or not the results will hold true in the fly. A specific point of confusion is that S2 cells are male cells that, according to modEncode data, express Fru (Flybase). Why then does S2-cell lysate produce Lola29F in absence of proteasome inhibitors? These results either confound the model or reflect a peculiarity of S2 cells that would require them to be validated for their suitability in this study.

A6. In the current study, S2 cells were used for systematic point-mutant analysis of ubiquitin-proteasome actions and pharmacological tests of proteasome enzyme inhibitors, two types of experiments that are feasible in cultured cells but difficult to carry out in flies *in vivo*. We performed western blot analysis with S2 lysates for Fru expression several times, but we did not detect Fru expression. Moreover, even if Fru is expressed in S2 cells (as modEncode reported), the S2-derived Fru is likely non-sex-specific FruCOM rather than FruM, because the S2 cells originated from non-neural cells (probably lymphocytes), while FruM is specifically expressed in neural cells of males. This might explain why Lola29 was degraded in S2 cells even though these cells were “male”; an ~101 a.a. stretch in the very N-terminus of FruM is lacking in FruCOM. Because we think that our experiments with S2 cells to deduce lysine residues critical for degradation control and to examine possible actions of enzyme inhibitors on degradation merit publishing, we have retained the relevant passages and figures from the original submission.

Q7. Furthermore, while the EMSA assays testing Lola29M/F *robo1* binding and inhibitory function

offer nice in vitro evidence, significantly stronger support could be provided with in vivo ChIP-qPCR using the same animals assayed for robo1 expression performed in Fig. 4l.

A7. Although we agree with the reviewer's view that ChIP-qPCR would be quite informative, it is technically demanding and difficult to complete within the time frame of this revision. We will endeavor to carry out ChIP-qPCR to test the in vivo binding of Lola29M/F on the robo1 promoter in a future experiment.

Q8. Some of the described experiments are vague regarding the methodologies used and their outcomes. For instance, the screen used to identify lola lacks a clear description of other positive hits (e.g. was robo1 found to be a suppressor of the GMR>FruBM eye phenotype?) and other genes that were identified. The mass spec analysis of the N-terminal Lola29M co-IP is similarly lacking in detail. Of particular importance, how was Cullin1 selected for further analysis, given that it was ranked 31 of the 121 interacting proteins identified, most of which appear to be housekeeping genes?

A8. We did not describe other genes identified as modifiers of the fru-induced eye phenotype, because they were previously listed in a table in our *Journal of Neuroscience* paper (Goto et al., 2011, J. Neurosci. 31, 5454-5459). We now refer to Goto et al. (2011) as a reference for other modifiers (line 84). The mass spec analysis was conducted with the aim of seeking clues for possible mediators of Lola29 N-terminal degradation, but it yielded only a few proteins, including Cullin1, that appeared to be promising candidate molecules for the mediation of Lola29M processing based on our knowledge of protein degradation. We have extended the description of our method for mass spec analysis in the Methods section (lines 786-797), and also clarified the passage in the Results section explaining the reason for this analysis—namely, to identify molecules involved in the proteolytic degradation of the Lola29 N-terminus (line 345).

Minor points

Q9. It should be demonstrated, if possible, that Cullin1 and Lola29F are expressed in mAL

A9. An anti-Cullin1 antibody labeled the mAL neurons. This observation is now described in lines 352-353, and an image of the labeled neurons is shown in Supplementary Figure 12. On the other hand, it is practically impossible to demonstrate Lola29F expression in mAL neurons because Lola29F is derived from Lola29M by N-terminal truncation, and thus the antibody that recognizes Lola29 inevitably detects both isoforms and is unable to distinguish one from the other. (Please note that Lola29M expression in mAL neurons was demonstrated in Figure 1k and l in the original submission).

Instead, we generated an mAL MARCM clone in which *Cul1* was knocked down, as this manipulation should increase the amount of Lola29M at the expense of Lola29F, and such a change in the protein amount might be detected by an antibody that recognizes the Lola N-terminal sequence that is present in Lola29M but absent in Lola29F. We compared the staining intensity of mAL clones expressing *Cul1* RNAi with the staining intensity of surrounding Lola-positive cells. We similarly compared the staining intensity of control mAL clones that did not express *Cul1* RNAi with other Lola-positive cells. As can be seen in the figure attached here, antibody staining was more intense in the clones with *Cul1* knockdown (middle panel) than the control clones (left-hand side panel), in keeping with the idea that Lola29F is produced in mAL neurons in female flies (Quantitative data are shown in the right-hand side panel).

[legend] A mAL clone with *Cull* knockdown (middle panel) was more intensely stained than a control clone (left-hand side panel) by anti-LolaCOM that recognizes the N-terminus of Lola (Quantified in the right-hand side panel. n=15 for both control and test clones.) Fluorescent intensity of a cell was measured using ImageJ (v1.52f, NIH) software according to the method described in McCloy et al. (2014). *Cell Cycle*, **13**, 1400-1412. Using ImageJ, the

outline of a GFP positive cell (where *Cull* was knocked down) was drawn, and area (Area) and integrated density (IntDen) in the area were measured, along with the mean fluorescence of the adjacent background reading (BKGD). The fluorescent intensity was calculated by the equation (a) = IntDen – (Area x BKGD). The fluorescent intensity (b) was similarly measured in a neighboring GFP negative cell (where *Cull* was not knocked down), and the relative fluorescent intensity was calculated by dividing (a) by (b). Box plots and statistical analyses (Mann-Whitney’s U-tests) were performed using GraphPad Prism 7 (*: p<0.05).

Q10. Since *lola* is broadly expressed, a nice demonstration of their model would be that Lola29F is present in Fru- neurons.

A10. We were unable to specifically mark Lola29F in neurons for the reason mentioned—i.e., because the available antibody that recognizes Lola29F inevitably labels also Lola29M. However, as also described in A9, we found an increase in the relative amount of Lola29M, the precursor of Lola29F, upon blocking its degradation by Cullin1 in fru-positive mAL neurons, which indeed express Lola29M (Figure 1k and l).

Q11. In the introduction the authors state: “Here we demonstrate that the male-biased Lola29M isoform forms a complex with FruBM to repress transcription from robo1 promoter;”, which suggests that they believe that the FruBM-Lola29M interaction is necessary for both proteins’ inhibitory roles and not merely for the protection of Lola29M from degradation. The model in figure 7 further supports this impression. This conclusion is unsupported by the evidence and the two proteins may both inhibit transcription independently from one another (supported by the observation that Lola29MK41R promotes ipsilateral neurite formation in females, independent of any FruBM product). The authors should clarify their model in the text and figure 7 to eliminate any potential confusion.

A11. Thank you very much for pointing out that the model presented in the original submission contains the unwarranted proposition that Lola29M and FruM cooperatively repress transcription from the robo1 promoter. We agree that whether FruM represses robo1 transcription independently from (in addition to) Lola29M remains an open question. We have modified the passage as follows (lines 45-52): “Interestingly, the *fru* gene product FruM represents another, BTB-zinc finger protein group, which includes a set of male-specific proteins (i.e., FruAM, FruBM and FruEM: nomenclature according to Ref. 9; Supplementary Fig. 1) that function to masculinize certain neurons¹⁰⁻¹² presumably via chromatin remodeling⁹. For example, FruM represses transcription from *roundabout1* (*robo1*), a negative regulator gene for neuritogenesis, thereby allowing male-specific neurite formation in males¹³. Here we demonstrate that the male-biased Lola29M isoform forms a complex with FruBM”. We have also modified our illustration of the model accordingly (Fig. 7).

Q12. It would be nice to see that Lola29MK41R overexpressing clones show reduced robo1 staining compared their non-clonal neighbors in females and that the reciprocal is true in Lola29F overexpressing clones in males.

A12. The level of endogenous Robo1 expression in mAL neurons is low, making it difficult to show its changes due to genetic manipulation. As a practical solution to this problem, we examined the CNS of wandering stage larvae, which gave us a reasonably strong Robo1 immunoreactivity.

Lola29MK41R overexpression via Elav-GAL4 resulted in a reduced Robo1 expression, as now shown in Supplementary Figure 10 and described in lines 279-281.

Reviewer #3 (Remarks to the Author):

Q13. Why did authors focus on Lola isoforms with exon29 among many others? Authors need to provide a rationale. Otherwise, authors may need to examine at least a few other Lola subtypes to evaluate sub-type specific role of Lola29M/F.

A13. We added the following sentence to describe our rationale for focusing on exon 29 (lines 100-104): “To obtain hints as to which of the Lola isoforms might have a role in the fru-dependent sexual differentiation, we examined possible effects of isoform-specific knockdown for isoforms 11, 17, 22, 26, 28 and 29, for which UAS-RNAi transgenic strains were publicly available, and found that isoforms 22 and 29 interfered with the sex-specific development of fru-expressing neurons (see below).”

Q14. Does Lola29M affect the stability of FruBM? Did authors evaluate the possibility that Lola29M may also increase the activity of FruBM by protecting FruM from degradation? Is it possible that knockdown of Lola29M reduce FruBM levels?

A14. At least N-terminally truncated FruBM has never been detected. We have carried out western blot analysis for FruBM with and without Lola29M knockdown as targeted by *fru-GAL4*, and found no difference in FruBM expression between the two conditions. A photograph of a representative western blot result is shown at left.

[Legend] Western blot analysis of protein extracts from the CNS of male pupae in which FruBM with a C-terminal HA tag was overexpressed via *fru-GAL4*. The presence or absence of the indicated transgenes is shown by the + or – marks above the image of a gel. The size and abundance of FruBM are not discernibly affected by Lola29 knockdown, suggesting that Lola29 has no role in the integrity of FruBM. α -Tubulin (α -Tub) was used as a loading control.

Q15. Can other FruM-types (type A, type E) also interact with Lola29M, at least in protein levels?

A15. We did not examine whether FruAM and FruEM interact with Lola29M because nothing is known about their roles in the male-specific neurite formation in mAL neurons. In contrast, FruBM has an established role in this process, with the known target gene *robo1* and the sequence in the *robo1* gene promoter that mediates its interactions with FruBM. Indeed, we are interested in possible interactions of Lola29M with FruAM and FruEM, and will conduct an experiment to test if this happens to be the case in the future.

Q16. Fig. 3c: Why *lola29m*[D1-300] cannot inhibit production of ipsilateral neurite completely? Is this because FruMs override Lola29F function or FruMs function independently (at least partially) from Lola29M? This result needs an explanation.

A16. Thank you for the thoughtful comment. We have added the following sentence in the revised manuscript (lines 210-213): “However, mAL neurons with the ipsilateral neurite were still produced, though at a reduced rate, in male flies with Lola29F-like overexpression, implying that an additional, Lola29F-resistant mechanism operates for the ipsilateral neurite formation.”

Q17. How does Lola29F (i.e., *Lola29M*[D1-300]) block binding of Lola29M on *robo1* promoter? Does inhibitory function of Lola29F require Zn-finger domain? Can Lola29F expression alone compromise the male courtship vigor as well?

A17. We presume that Lola29F-like competes with Lola29M for some cofactors required for the repressor function of Lola29M, and this possibility needs to be experimentally tested. However, the elucidation of the mechanism whereby Lola29F inhibits Lola29M function would require significant time and labor, and we feel this subject is beyond the scope of the current paper.

Q18. Fig 4h, 4i: Overexpression of *lola29m*[K41R] or *lola29m*[D1-300] did not fully restore male courtship vigor or female mating receptivity either. Is this because *lola-exon29-RNAi* also reduces expression of exogenous *lola29m*[K41R] or *lola29m*[D1-300]. If so, how do authors rule out the possibility that *lola-exon29-RNAi* suppress exogenous *lola29m*[K41R] and *lola29m*[D1-300] in different extents?

A18. To incorporate the reviewer’s opinion, we have added the following passage (lines 263-267): “We note, however, that neither Lola29M nor Lola29F-like was able to resume the mating activity to the normal level in male or female flies expressing *lola-exon29 RNAi*. The partial rescue of mating

behavior by overexpressed Lola29M or Lola29F-like might suggest that mRNAs derived from the transgenes were also targeted by the *lola-exon29 RNAi* to a certain extent.”

Q19. Line 315: How does 26S proteasome degrade Lola29M only partially but yet precisely? Does 26S proteasome have some sequence specificity? Is there any known example for such specific degradation? If so, the discussion should include such case.

A19. There is a precedent case that involved partial and specific degradation by 26S proteasome. We added the following sentence to explain this point (lines 428-434): “In this context, it would be worth mentioning that Cul1 and Cul3 play contrasting roles in the regulation of *hedgehog* (*hh*) signaling; in the absence of Hh, Cul1 is recruited to the hh downstream zinc finger protein Cubitus interruptus (Ci155), which undergoes, as a consequence, partial degradation to yield the transcriptional repressor Ci75, whereas upon the Cul3 recruitment in the presence of Hh, Ci155 is completely degraded³¹. It remains to be determined how Cul1 is recruited to Lola29M for its partial degradation in fru-positive neurons.”

Q20. Does the loss of DR1 affect the binding of FruBM on FROS site? Previous work from the authors’ group (Ref# 19) may answer this question. Authors need to clarify this point in the discussion.

A20. Our previous result showed that loss of DR1 does not impair FruBM binding to the *robo1* promoter (Ito et al., 2016). In this revision, we conducted an additional behavioral experiment with flies that lack DR1, which revealed that these males prematurely switch from one wing to the other when they vibrate wings for courtship song generation. Such precocious wing switching has been interpreted as a result of malformation of the ipsilateral neurite of mAL neurons, possibly due to the incomplete repression by FruBM of *robo1* transcription (Ito et al., 2016). We have expanded the Results section to include this finding by adding Supplementary Figure 11, Supplementary movie 1 and a new passage (lines 321-331): “DR1 lies just outside FROS and, therefore, is dispensable for FruBM binding to the *robo1* promoter¹³. Remarkably, however, male flies homozygous for *robo1*[Δ4], a *robo1* mutant carrying a 10 bp deletion that removes DR1, exhibited precocious wing switching during courtship (Supplementary Figure 11), which represents a behavioral phenotype uniquely observed in flies with defects in the male-specific neurite formation of mAL neurons¹³. For example, *robo1*[Δ1] and *robo1*[Δ2] mutations that delete a part of the core palindrome Pal1 in FROS dominantly induce the precocious wing switching in male flies¹³. The *robo1*[Δ4] mutation was recessive in inducing the precocious wing switching and had no dominant effect (Supplementary Figure 11),

suggesting that Lola29M bound to DR1 plays a distinct role that is needed for FruBM to fully repress robo1 transcription.”

Minor comments

Q21. Line 25-27: Authors need to elaborate this point in the discussion or the introduction

A21. We have added the following sentence to the last paragraph in the Introduction to convey the importance of our work in society (lines 61-69): “In humans, sexually biased incidence of certain neurological disorders has been well recognized, yet the origin of such sex differences remains largely an enigma. For instance, the male-to-female incidence ratios have been reported to vary from 1.37 to 3.7 in Parkinson’s disease, the etiology of which likely involves mitochondrial dysfunction due to defects in the proteasomal degradation system¹⁴. Our finding in *Drosophila* that the neuronal sex-type specification involves proteasomal protein processing will shed light on the hitherto unknown mechanistic link among posttranslational protein modification, neural sex differentiation and complex neurobehavioral traits under normal and disordered conditions.”

Q22. Line 38: How do authors produce Supplemental Fig.1? Where does the data (about splicing isoforms) of lola come from? Please clarify the sources.

A22. In the legend of Supplementary Figure 1 of the revised manuscript, we have cited references for the splicing isoforms: “The exon-intron organization and possible splicing patterns are drawn based on Goeke et al. (2003) and Ohsako et al. (2003) for lola and Billeter et al. (2006) and Ito et al. (2012) for fru.”

Q23. Line 76. … well established: Add reference.

A23. We have cited Ginger et al. (1994) as a reference in the revised manuscript (line 88).

Q24. Line 87. … identified to date: Add reference.

A24. In the revised manuscript, we have cited Dinges et al. (2017) as the most recent paper on lola (line 100).

Q25. Line 91-92. The isoform … as type-Q: I cannot find this information in Ref14. Please clarify this.

A25. In this revision, we have cited Dinges et al. (2017) in place of “Red14” as a reference in which

type-Q is referred to (line 109).

Q26. Ref14 reported the *in vivo* evidence indicating that *lola* interacts with *robo*. Authors need to introduce or discuss the relevant previous observations with sufficient details.

A26. We added the following passage in the Results section to describe the known genetic interaction between *lola* and *robo1* (lines 273-275): “Of note, Crowner et al.¹⁸ have reported that a *robo1* mutant copy dominantly enhances the axon misrouting phenotype in a weak hypomorph of *lola*, *lola*[ORE120], which barely manifests this phenotype on its own.”

Q27. Line 96: Where does *Lola29F* occur in the female brain? Does it occur in fru neurons? Need to show the actual data.

A27. Because *Lola29F* is derived from *Lola29M*, the anti-*lola*-exon29 antibody cannot differentiate one from the other. We do not have any means to selectively label *Lola29F*. However, one can anticipate that an antibody that recognizes the N-terminus of *Lola* would detect an increase in the amount of *LolaM* contained in neurons where N-terminal truncation is blocked to prevent the production of *Lola29F* from *Lola29M* by proteolysis. Based on this consideration, we compared the intensity of immunoreactivity to the anti-*Lola* antibody between the mAL clones with and without *Cul1* RNAi expression, while the staining intensity of *Lola*-expressing neurons outside the clone was used as a reference for the staining variations for technical reasons. We found that the mAL clones subjected to *Cul1* knockdown were more strongly labeled than their control counterparts (see the figure appended to A9 above). This observation was consistent with the idea that *Lola29F* is present in mAL neurons as a truncation product of *Lola29M*, provided that *Cul1* can act on it.

Q28. Line 124: how do authors predict the truncation position between residues 250 and 300?

A28. The apparent size of *Lola29F* on the western blot was empirically close to that of *Lola29M* with a deletion of 250-300 residues, and the antibody recognizing the N-terminal tag failed to detect *Lola29F*. We therefore suspected that the truncation occurs somewhere between a.a. 250 and a.a. 300. Subsequent Edman degradation analysis unequivocally demonstrated that a.a. 1-263 of *Lola29M* were removed in *Lola29F*.

Q29. Lines 125, 158: No mass spectrometry data here.

A29. We have deleted “and mass spectrometry” in the relevant passages of the revised manuscript.

Q30. Line 173: Need to show that Lola29M is indeed ubiquitinated.

A30. Our additional experiment provided data indicating that Lola29M is ubiquitinated. We have therefore added the following passage and a new supplementary figure to present the new results (lines 190-195): “Polyubiquitination of ubiquitin at lysine 48 (K48) is known to direct substrate proteins to proteasome-mediated degradation²⁴. We found that a K48-linkage-specific polyubiquitin antibody coimmunoprecipitates Lola29M/F in lysates from S2 cells transfected with *lola29m*, and fruBM cotransfection markedly diminishes the yield of immunoprecipitates (Supplementary Figure 8). This result is taken as evidence that Lola29M is indeed ubiquitinated in the absence of FruM.”

Q31. Line 209. “;female counterparts terminate in the contralateral neuromere”;
The male fibers seem to terminate in the contralateral neuromere, but female ones do not.

A31. Thank you for pointing out the error in the original manuscript. We have replaced “contralateral” with “ipsilateral” in the revised manuscript (lines 234-235).

Q32. Line 304: difficulty in following this logic. Is this because Cullin1 cannot access the full-length Lola29M bound with FruBM?

A32. Our assumption was that partially degraded proteins would be more susceptible to proteasome-mediated degradation. To imitate such conditions, a.a. 1 – a.a. 150 of Lola29M were deleted.

Q33. Line 396-400: Elaborate this point with more specific examples.

A33. We have added the following sentence to enhance the final part of the Discussion (lines 455-465): “Moreover, there is accumulating evidence that the association of BTB proteins with ubiquitin ligases plays multilayered regulatory roles in developmental decisions. For example, Germ cell-less (Gcl), a BTB protein conserved from *C. elegans* to humans with a transcriptional repressor activity³³, plays a key role in the soma-germ fate switch via a non-transcriptional mechanism. When complexed with Cul3, Gcl exits the nucleus to degrade the somatic fate determinant Torso and promotes the primordial germ cell fate in *Drosophila*³⁵. In vertebrates, the Cul3-KBTB18 complex switches the translational program so as to specify the neural crest fate³⁶. The present study further expanded the roles of the ubiquitin proteasome system—namely, the system was shown to function in the specification of sex-types of neurons via regulated proteolysis of a transcription factor that functions in the sex-determination molecular machinery.”

Q34. Line 875: NS is absent in Fig. 2.

A34. We deleted “NS: Non significant.” (lines 1001-1002)

Q35. Line 955: P1 and P2 are absent in Fig 5.

A35. We deleted the passage “P1 and P2 indicate promoters for robo1-RA and robo1-RC transcripts.”

Q36. Line 958: It should read as DeltaDR1. $\Delta DR1$; is missing.

A36. We corrected ΔDR to $\Delta DR1$.

Q37. Fig. 1d: There is virtually no signal of Lola29F in CS. How could authors normalize data with zero value of CS?

A37. We used ImageJ to quantify the intensity of signals on the gel. The signal intensity for CS males represents the value at the position where the Lola29F band is occupied in CS females, which in practical terms corresponds to the value for background noise.

Q38. Fig. 1l: It is not clear which cells are positive for GFP in Fig. 1l.

Please label them.

A38. We labeled the GFP-positive cells with arrows.

Q39. Fig. 2h. What does the shield mark indicate? Specify it.

A39. We deleted the shield mark.

Q40. Fig 4l: Which sex was used to generate this figure?

A40. We changed the phrase “with fly extracts” to “with male fly extracts” in order to clarify that we used males (line 276 of the main text).

Q41. Fig 5b, c, e, d, h: Fold suppression is difficult to comprehend. Provide formula how to produce it either in the figure legend or Materials and Methods.

A41. We added a sentence in the legend to explain how we calculated “fold suppression” as follows: “To normalize the transfection efficiency, raw values for firefly luciferase activity were divided by raw *Renilla* luciferase activity values. The values representing fold suppression (x) were calculated by dividing the normalized luciferase-activity values (lucA) in cells cotransfected with the empty vector

pact-MCS by those (lucB) in cells cotransfected with pact-FLAG-fruBM (x=lucA/lucB). All experiments were carried out in triplicate, and data are presented as the fold suppression in mean (\pm SEM) relative luciferase activity.” (lines 721-727)

Supplemental data

Q42. Line 44. right-hand side) in (c-g); There is no scale bar in Supplemental Fig2 d, e, f, g.

A42. The scale bar shown in panel c applies to panels c-g, as now described in the Supplementary Figure 2 legend.

Q43. Supplemental Fig. 4: need to include double labeling of anti-Lola29 and anti-Pros.

A43. This series of experiments was carried out to demonstrate that FruM is selectively expressed in differentiating neurons, where Lola29 is also expressed. Pros was used to distinguish differentiating neurons from neuroblasts, which do not express Pros. We do not see the necessity of conducting double labeling of cells with Pros and Lola29, because we have shown the coexpression of Lola29 and FruM in panels j-l of the same figure.

Q44. Supplemental Fig. 5c: What 5'RACE primer was used.

A44. All primers were described in Table S3. For 5' RACE, the forward primer was a mixture of oligo named “5' RACE primer (F)”, which was used in conjunction with the reverse primer named “Primer 1 (R)”, a gene-specific primer. For 3' RACE, the forward primer was the gene-specific “Primer 2 (F)” in our terminology, which was used in conjunction with the “3' RACE primer (R)”, a mixture of oligo. We have revised the relevant passage (lines 697-700) in the Methods to avoid any confusion.

Q45. Supplemental Fig 5d: Show male and female data separately.

A45. The male product Lola29M has the conserved N-terminal sequence common across the majority of Lola isoforms, which are encoded by mRNAs transcribed by one of the 5' four distal promoters, P1-P4 (which respectively yield sequences from the non-coding exon1, exon2, exon3 and exon4 in 5'RACE). In contrast, the female product Lola29F lacks the common N-terminal sequence, which raises the possibility that the female-specific isoform is derived from an mRNA transcribed from a local promoter situated in a downstream intron. We conducted the 5'RACE to determine whether this is the case. The results shown in Supplementary Figure 5d demonstrated that the female-type isoform

Lola29F is encoded by an mRNA transcribed by either P2 or P3, i.e., a conventional distal promoter, and thus we can exclude the possibility that the local promoter is involved. In view of the rationale described above, we do not think that the experiment with male extracts is needed.

Q46. Line 93-94. No need to use capital letters.

A46. We changed this to lowercase.

REVIEWERS' COMMENTS:

Reviewer #1 (Remarks to the Author):

I am satisfied with the revisions provided by the authors in the new version of the manuscript.

Reviewer #2 (Remarks to the Author):

Overall, I appreciate the authors' efforts to address our concerns. However, several important points remain and should at least be noted with textual changes.

1) Specifically, I remain concerned that samples sizes were not consistent across or even within experiments. Since clonal analysis is inherently stochastic the best practice is to determine a standard sample size based on the observed distribution of controls. The authors' assertion that the p-value would only get smaller if the n was increased is only true if the sample size chosen is large enough to accurately sample the full distribution of the data.

2) Regarding the strengths of the mutant phenotypes-- I appreciate now that the authors' goal for these experiments was to demonstrate the masculinizing effect of Lola29M overexpression as qualitatively as possible, i.e. in fru-null animals and females, which they do effectively. However, I'm confused by the assertion: "Please note that the masculinizer effect of overexpressed Lola29M cannot be demonstrated in fru[+] males, because these males develop a full-sized ipsilateral neurite even without any manipulation, and thus there is no room for further masculinization". Are the two subclasses (+/- ipsilateral neurite) described for fruNP21/+ male mAL neurons not representative of the mAL neurons found in WT fru+ males? If so, it needs to be clarified in the text that fruNP21 heterozygous animals used throughout this study are haploinsufficient. If this is not what the authors meant then I maintain that it would be interesting to know the effects of Lola29M overexpression in fruNP21/+ males. The authors report that in this background 12.5-17.6% of mAL neurons lack the ipsilateral neurite, is this due to insufficient Lola29M activity? This result would seem relevant to the authors' model and our understanding of the heterogeneity that exist in mAL neurons in males.

3) I appreciate the authors' rationale for using S2 cells due to their ease of use. It would still help to include the western blots of S2 cell lysate with FruCOM and FruM antibodies in the supplemental data since expression patterns in cell culture do not always match the in vivo expectations.

There is a typo in line 275, RT-RCR should read RT-PCR.

Reviewer #3 (Remarks to the Author):

I have no further comment. It is ready for publication.

Point-by-point replies to the reviewer's comments

Reviewer #2 (Remarks to the Author):

Q1: Specifically, I remain concerned that samples sizes were not consistent across or even within experiments. Since clonal analysis is inherently stochastic the best practice is to determine a standard sample size based on the observed distribution of controls. The authors' assertion that the p-value would only get smaller if the n was increased is only true if the sample size chosen is large enough to accurately sample the full distribution of the data.

A1: To draw readers' attention to potential problems arising from the small number of samples in some of our clone experiments, we have added the following sentence (lines 653 – 670), in which the results of power analysis for relevant clone data are described: “The number of clones examined was different from experiment to experiment as a consequence of the stochastic nature of mosaic generation. This might complicate interpretations of data, particularly in the experiment shown in Figs 3c, 6c and 6f, where the proportion of neurons with the ipsilateral neurite was quantitatively compared. We therefore performed a post hoc power analysis of these data with the aid of Webpower⁴³, which yielded the following results: Power=0.7194, $\alpha=0.05$ for Fig. 3c; power=0.3095, $\alpha=0.05$ for Fig. 6c; power=0.9774, $\alpha=0.05$ (*fru*^{NP21/+} vs. *fru*^{NP21/UAS-Cul1 RNAi}) and power=0.8209, $\alpha=0.05$ (*fru*^{NP21/+} vs. *fru*^{NP21/UAS-Pros- β 5 RNAi}) for Fig. 6. Here, the sample number is considered to be large enough when it gives a power value larger than 0.7. The data shown in Figs. 3c and 6f fulfilled this criterion but those in Fig. 6c did not. Our calculations indicate that power values over 0.7 would be obtained if 23 or more samples were subjected to analysis in the experiment shown in Fig. 6c (*cf.* 8-15, the current sample numbers). However, the flies with the genotype *robo1*^{GA285/+}; *fru*^{NP21/UAS-lola29M^{K41R}} were extremely difficult to obtain as most of the individuals died before eclosion. Because the statistical difference at P<0.01 found in this set of data stands regardless of whether the power exceeds 0.7 or not, we consider that the

result in Fig. 6c supports our conclusion that *robo1* and *lola* (Lola29M) interact *in vivo*.”

Q2: Regarding the strengths of the mutant phenotypes-- I appreciate now that the authors' goal for these experiments was to demonstrate the masculinizing effect of Lola29M overexpression as qualitatively as possible, i.e. in *fru*-null animals and females, which they do effectively. However, I'm confused by the assertion: “Please note that the masculinizer effect of overexpressed Lola29M cannot be demonstrated in *fru*[+] males, because these males develop a full-sized ipsilateral neurite even without any manipulation, and thus there is no room for further masculinization”. Are the two subclasses (+/- ipsilateral neurite) described for *fru*NP21/+ male mAL neurons not representative of the mAL neurons found in WT *fru*+ males? If so, it needs to be clarified in the text that *fru*NP21 heterozygous animals used throughout this study are haploinsufficient. If this is not what the authors meant then I maintain that it would be interesting to know the effects of Lola29M overexpression in *fru*NP21/+ males. The authors report that in this background 12.5-17.6% of mAL neurons lack the ipsilateral neurite, is this due to insufficient Lola29M activity? This result would seem relevant to the authors' model and our understanding of the heterogeneity that exist in mAL neurons in males.

A2: *fru*[NP21] heterozygotes do not exhibit haploinsufficiency. When mAL neurons in the male brain were labelled with the aid of certain flylight GAL4 lines (they are wild type for the *fru* locus), the neurons without the ipsilateral neurite as well as the neurons with the ipsilateral neurite were visualized (e.g., Costa et al., 2016, Neuron 91, 293-311). We suspect that some of mAL neurons that lack the ipsilateral neurite are “intrinsically” devoid of this neurite irrespective of whether Lola29M is there or not, despite a lack of evidence for or against this idea. Based on these considerations, we have added the following sentence to the text (lines 469 – 474): “The GAL4 driver *fru*^{NP21} used in this study is a recessive allele of *fru* induced by a P-element insertion into the *fru* second intron (Supplementary Fig. 1a), which exhibited no discernible defect in neurite

structures. A subset of mAL neurons lacked the ipsilateral neurite in *fru*^{NP21/+} heterozygous males, as were the case in males of some flylight lines³⁷, which carried driver GAL4 insertions at genomic sites unrelated to the *fru* locus³.”

Q3: I appreciate the authors' rationale for using S2 cells due to their ease of use. It would still help to include the western blots of S2 cell lysate with FruCOM and FruM antibodies in the supplemental data since expression patterns in cell culture do not always match the *in vivo* expectations.

A3: We have added a supplementary figure (Supplementary Fig. 14), which shows western blot analysis of S2 cell lysate probed with an anti-FruM antibody. The FruM (more specifically FruBM) was detected only upon transfection of cells with the *fru*BM-encoding sequence, demonstrating that FruM was not expressed endogenously in S2 cells. We were unable to conduct a corresponding experiment with the anti-FruCOM antibody, as this antibody did not work in western blotting. We have added the following passage (lines 563 – 570) to clarify that S2 cells do not express FruM: “The endogenously expressed Fru proteins may affect transfection assays in S2 cells. We performed western blot analysis with S2 lysates for FruM expression several times, detecting no endogenous FruM expression (Supplementary Fig. 14). Even if Fru is expressed in S2 cells, the S2-derived Fru is likely non-sex-specific FruCOM rather than FruM, because the S2 cells originated from non-neural cells (probably lymphocytes), while FruM is specifically expressed in neural cells of males. We were unable to determine whether FruCOM is expressed in S2 cells due to the lack of an anti-FruCOM antibody that works in western blotting.”

Q4: There is a typo in line 275, RT-RCR should read RT-PCR.

A4: The typographic error has been corrected.

Reviewer #3 (Remarks to the Author):

I have no further comment. It is ready for publication.